# Compartments in medulloblastoma with extensive nodularity are connected through differentiation along the granular precursor lineage

David R. Ghasemi[1,2,3,19], Konstantin Okonechnikov[1,2,19], Anne Rademacher[4], Stephan Tirier[4,18], Kendra K. Maass[1,2,3], Hanna Schumacher[5], Piyush Joshi[1,2], Maxwell P. Gold[6], Julia Sundheimer[1,2,7], Britta Statz[1,2], Ahmet S. Rifaioglu[5,8], Katharina Bauer[9], Sabrina Schumacher[4], Michele Bortolomeazzi[9], Felice Giangaspero[10,11], Kati J. Ernst[1,12], Steven C. Clifford[13], Julio Saez-Rodriguez[5], David T. W. Jones[1,12], Daisuke Kawauchi[14], Ernest Fraenkel[6,15], Jan-Philipp Mallm[9], Karsten Rippe[4], Andrey Korshunov[1,16,17,20] ✉, Stefan M. Pfister[1,2,3,20] ✉ & Kristian W. Pajtler[1,2,3,20] ✉

Medulloblastomas with extensive nodularity are cerebellar tumors characterized by two distinct compartments and variable disease progression. The mechanisms governing the balance between proliferation and differentiation in MBEN remain poorly understood. Here, we employ a multi-modal single cell transcriptome analysis to dissect this process. In the internodular compartment, we identify proliferating cerebellar granular neuronal precursor-like malignant cells, along with stromal, vascular, and immune cells. In contrast, the nodular compartment comprises postmitotic, neuronally differentiated malignant cells. Both compartments are connected through an intermediate cell stage resembling actively migrating CGNPs. Notably, we also discover astrocytic-like malignant cells, found in proximity to migrating and differentiated cells at the transition zone between the two compartments. Our study sheds light on the spatial tissue organization and its link to the developmental trajectory, resulting in a more benign tumor phenotype. This integrative approach holds promise to explore intercompartmental interactions in other cancers with varying histology.

Medulloblastoma (MB) is the most common embryonal brain tumor of childhood and accounts for a significant proportion of both morbidity and mortality in this age group[1,2]. Traditionally, MB has been stratified into four histopathological subtypes based on histological appearance: Classic, Large Cell/Anaplastic, Desmoplastic/Nodular (DNMB) and MB with Extensive Nodularity (MBEN)[3,4]. Additionally, in the last decade, four molecular groups (WNT, SHH, Group 3, and Group 4) together with various subgroups have been defined and now generally replace the classic histopathological stratification in the 2021 WHO classification of central nervous system (CNS) tumors[5–14].

A full list of affiliations appears at the end of the paper.  ✉e-mail: andrey.korshunov@med.uni-heidelberg.de; s.pfister@kitz-heidelberg.de; k.pajtler@kitz-heidelberg.de

MBEN is a unique histological type of MB that mainly arises in very young children and infants. These tumors exclusively fall into the molecular SHH-group and have been shown to frequently harbor mutations in SHH-pathway genes, for instance, *PTCH1* or *SUFU*[4,15]. Their distinctive, bicompartmental histological appearance is characteristic and differentiates them from other MBs: large nodular conglomerates of postmitotic, differentiated, neurocytic tumor cells are surrounded by internodular zones of highly proliferative, poorly differentiated cells[15–17]. Furthermore, MBEN show a distinctive, grapevine-like neuroradiological appearance[16–18]. MBEN are generally associated with good prognosis and are therefore considered low-risk in comparison to other MB variants. However, relapse and disease progression can occur and may require intensified therapy, resulting in potentially life-long, severe side effects in these very young children[15,19–22]. Interestingly, case studies reported that MBEN may mature and differentiate into benign ganglion cell tumors[23,24]. These findings have led to speculations whether underlying biological differentiation programs may exist in these tumors that could result in a loss of malignant potential[16]. This hypothesis is underlined by the fact that MBEN are defined by the upregulation of cellular pathways that are important in cerebellar neuronal differentiation, for instance synaptic transmission, glutamatergic signaling and calcium homeostasis[25]. In the past, it has been disputed whether DNMB and MBEN represent two biologically distinct MB variants. We previously showed that these histological variants can be reliably distinguished based on their transcriptomic profiles, despite harboring similar SHH-associated epigenetic signatures[25].

Whereas clinical characteristics of MBEN are well established, the biology underlying its distinctive histological appearance and varying disease aggressiveness remains largely unknown. It is especially unclear if and how the two histological compartments of MBEN are interconnected. Several studies have postulated that the cells of origin of SHH-MB are cerebellar granule neuronal precursors (CGNP) in the external granular layer (EGL) of the developing cerebellum. The large differences in biological and clinical features within the molecular group of SHH-MB indicate that tumor formation may depend on distinctive spatial and temporal circumstances[26–30].

In this study, we use an integrated multi-modal approach that includes three complementary spatial transcriptomics methods. We demonstrate that MBEN mimics the development of CGNPs into granule neurons and that the bi-compartmental histology of MBEN represents distinct differentiation cell states that are connected through a direct developmental trajectory. Furthermore, we identify a subset of tumor cells that cluster together with non-malignant astrocytes and show an astroglial phenotype. Overall, our findings indicate that MBEN could be understood as a disease of the developing cerebellum and serve as an explanation why these tumors are almost entirely restricted to infancy.

## Results

### Molecular and clinical characterization of the MBEN cohort

In order to study the genetic basis of MBEN in a comprehensive way, we applied a multimodal set of complementary methods, including single nucleus RNA-sequencing (snRNA-seq) and spatial transcriptomics, to a cohort of nine fresh frozen MBEN samples (Fig. 1a, Supplementary Data 1). DNA methylation-based clustering of bulk methylomes with a reference cohort spanning all major molecular MB-groups confirmed that all cases belonged to the infant SHH-MB group (SHH-1: $n = 7$, SHH-2: $n = 2$) (Fig. 1b)[9]. DNA sequencing revealed genetic alterations of the SHH-pathway in six tumors (*PTCH1*: 1/9, *SMO*: 2/9, *SUFU*: 3/9) (Fig. 1c). As reported previously for MBENs, copy number variations (CNVs) were only detected infrequently (Supplementary Fig. 1). MBEN-histology was confirmed by central pathological review in all cases (A.K.) (Fig. 1d). All tumors were located in the cerebellum (Fig. 1e). At the time of diagnosis, disease stage was M0 in 7/9 cases, with two patients showing metastatic disease (M2: 1/9, M3: 1/9)

(Supplementary Fig. 2a). The median age of children included in this cohort was $2 \pm 0.70$ years and 7/9 patients were male (Supplementary Fig. 2b, c). In accordance with earlier studies, the clinical outcome in the presented cohort were overall favorable. However, a total of three out of nine children experienced relapses of their disease (Supplementary Fig. 2d, e).

### A subset of tumor cells is transcriptionally similar to non-malignant astrocytes

To investigate intra- and intertumoral transcriptional heterogeneity within MBEN, we isolated nuclei from fresh frozen material and subsequently applied two complementary methods of snRNA-seq (see methods), namely the 10X V2 3'- ($n = 9$) and the SMARTseq V2.5-protocols ($n = 6$). After initial quality control (Supplementary Fig. 3a–g), 28,132 and 1526 nuclei were used for downstream analyses, respectively. In order to ensure that snRNA-seq datasets faithfully represented the respective tumor, a fingerprint analysis using bulk RNA-sequencing profiles was conducted (Supplementary Fig. 4a). Furthermore, comparative clustering revealed concordant results between both methods, confirming that both techniques covered the same cell types (Supplementary Fig. 4b). Subsequently, we integrated both datasets to arrive at an integrated snRNAseq dataset resulting in 29,658 cells for further in-depth analysis (Fig. 2a, Supplementary Fig. 4c, d)[31]. Non-malignant cells were identified based on established marker genes or recently published single cell atlases and comparison to a reference dataset of the fetal cerebellum, which we recently published[32–34]. This approach resulted in the identification of astrocytic cells, oligodendrocytes/oligodendral precursors, monocytes/endothelial and fibroblast/ perivascular cells (Fig. 2b, Supplementary Figs. 3e–g, 4e, f). Apart from monocytes, no significant immunological infiltration, for instance by T- or NK-cells, was observed, confirming earlier studies that described MB as an immunologically cold tumor[35]. Surprisingly, analyzing genome-wide copy number variations (CNV) per individual cluster using inferCNV revealed a clear CNV-signature related to one case (MB295) in the astrocytic cluster (Fig. 2c)[36]. Since SMARTseq V2.5 generates whole transcriptome reads that can be used for the analysis of single nucleotide variants (SNVs), we investigated whether we could identify cells with both an astrocytic phenotype and pathogenic SNVs. Strikingly, a small number of astrocytic cells ($n = 3$) from sample MB266 indeed harbored the same mutation within the SHH pathway gene *SMO* as detected using bulk sequencing, further confirming our initial findings that a fraction of astrocytic cells in MBEN is derived from malignant precursors (Fig. 2d–f). In order to identify which astrocytic cells were malignant and non-malignant, respectively, we analyzed both snRNA-seq datasets separately and without correcting for patient-related batch effects. We hypothesized that malignant cells would cluster according to patient, while non-malignant cells would congregate in one cluster, as observed in other single cell studies[37]. Using this approach, we could indeed separate malignant from non-malignant cells within the astrocytic cluster, which partly clustered in mixed, non-malignant, and partly in malignant, patient-specific clusters (Fig. 2g, Supplementary Fig. 4g–j). Whereas the number of malignant astrocytic cells varied by sample, they were identified in every case (Supplementary Fig. 4g). Based on these findings, we hypothesized that the subpopulation of MBEN cells with astrocytic features represent astrocytic-like cells, which is supported by experimental studies that indicate that murine MB cells can transdifferentiate into tumor-derived astrocytes[38–40].

Taken together, we found that besides malignant cell subpopulations MBEN contains various non-malignant cells with glial, monocytic, vascular and endothelial transcriptional signatures. Interestingly, snRNA-seq using two complementary methods revealed a distinct subpopulation of tumor cells that correspond to astrocytic-like cells.

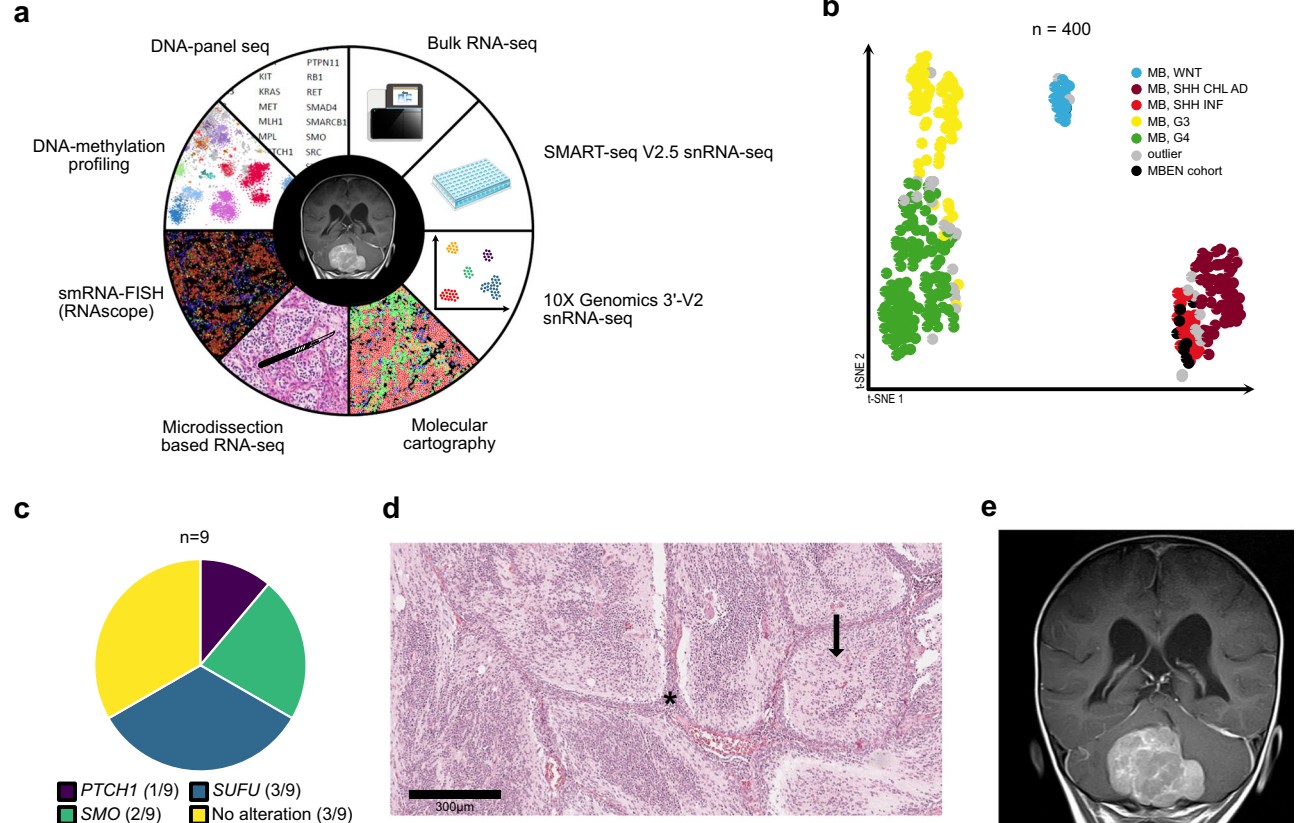

**Fig. 1 | Method spectrum and molecular features of the MBEN patient cohort. a** Visual summary of all methods that were applied to investigate the MBEN cohort. **b** t-SNE depicting a clustering of the MBEN cohort (*n* = 9) with a reference cohort of 391 molecularly characterized MBs. All nine samples clustered with the SHH-subtype. **c** Pie chart summarizing the alterations in the SHH-pathway that were detected using DNA panel seq. **d** H.E. staining showing the two characteristic histological components of MBEN. The black asterisk marks the internodular component, whereas the black arrow points towards a tumor nodule. All nine cases were histologically diagnosed as MBEN. Scale bar = 300 μm. **e** Representative coronal MRI image in an MBEN patient showing a large cerebellar mass with grapevine-like appearance. In order to design Fig. 1a, the two images "genomesequencer9" (by DBCLS, licensed under https://creativecommons.org/licenses/by/4.0/), and "multiwell-plate-3d" (by Servier, licensed under https://creativecommons.org/licenses/by/3.0/) from the database https://bioicons.com were used. Source data are provided as a Source Data file.

## Cell stages in MBEN recapitulate cerebellar granular development

In order to study MBEN cells in greater detail, non-malignant cell populations, including non-malignant astrocytes, were excluded from further tumor cell analysis. Re-analysis restricted to malignant cells revealed five distinct clusters and one additional one, which was entirely driven by upregulation of heat shock- and ribosomal-associated genes, indicating an artificial effect most likely induced by cellular stress associated with sampling (Fig. 3a). Only one cluster was actively proliferating as indicated by S- and G2M-signatures (Fig. 3b). The different cell populations showed distinct expression patterns matching those of individual CGNP developmental stages. Two clusters, of which one was proliferating, showed upregulation of marker genes of early CGNPs in the EGL (e.g., *BARHLH1, ZIC1, ZIC3, PTPRK*) and SHH-pathway members (*PTCH1, SMO, HHIP, GLI1, GLI2*). Additionally, the proliferating cluster expressed genes that are implicated in epigenetic regulation and chromatin remodeling (e.g., *EZH2, SUZ12, CTCF, YY1, RAD21*) (Fig. 3c, Supplementary Fig. 5d, Supplementary Data 2)[41–45]. These genes have been shown to be involved both in the orchestration of cerebellar development and the oncogenesis of numerous cancer types[45,46]. Clusters three and four were characterized by the expression of markers of intermediately differentiated, migrating (*GRIN2B, CNTN2, ASTN1, SEMA6A*) and postmitotic, differentiated CGNPs (*GABRA1, GABRA6, GRIN2C*), respectively (Fig. 3c, Supplementary Fig. 5e–i)[41,47–49]. A fifth cluster was formed by astrocytic-like cells.

Notably, we observed that markers associated with astrocyte-immune cell interactions in murine models (e.g., *IGF1, IL4*) did not show any specificity in human tumors, being spread across all cell types in MBEN samples[38]. In addition, astrocytic-like cells showed upregulation of stromal and of early cell stage markers (*LAMA2, SOX2, SOX9*) as well as SHH-pathway members (e.g., *HHIP, BOC, GLI2*) (Fig. 3c, Supplementary Fig. 5a–c), but no significant proliferative activity. Pseudotemporal ordering using Slingshot[50] revealed a continuous lineage starting from the proliferating, early CGNP-like cell states, spanning the intermediate ones and finally congregating in the postmitotic, neuronally differentiated cells (Fig. 3d). This lineage resembled physiologic differentiation of CGNPs into normal granular neurons during cerebellar development and was confirmed in an independent cohort of MBEN cases in an accompanying study by Gold et al. Astrocytic-like cells branched off early in the trajectory and prior to the appearance of markers of cell migration (Fig. 3d). In addition, we used monocle2 as an additional pseudotime ordering tool on sample MB295, which included a large fraction of astrocytic-like malignant cells. The obtained trajectory strongly reflected the main pattern that we observed from Slingshot, with early CGNP-like cells at the start, and neuronally differentiated cells at the end of the trajectory (Supplementary Fig. 5j–y). Interestingly, astrocytic-like cells demonstrated a slightly different location within the trajectory as compared to the cluster-based approach using slingshot and were found at the earliest time point prior to the clusters of early, CGNP-like cells. (Supplementary Fig. 5j, o).

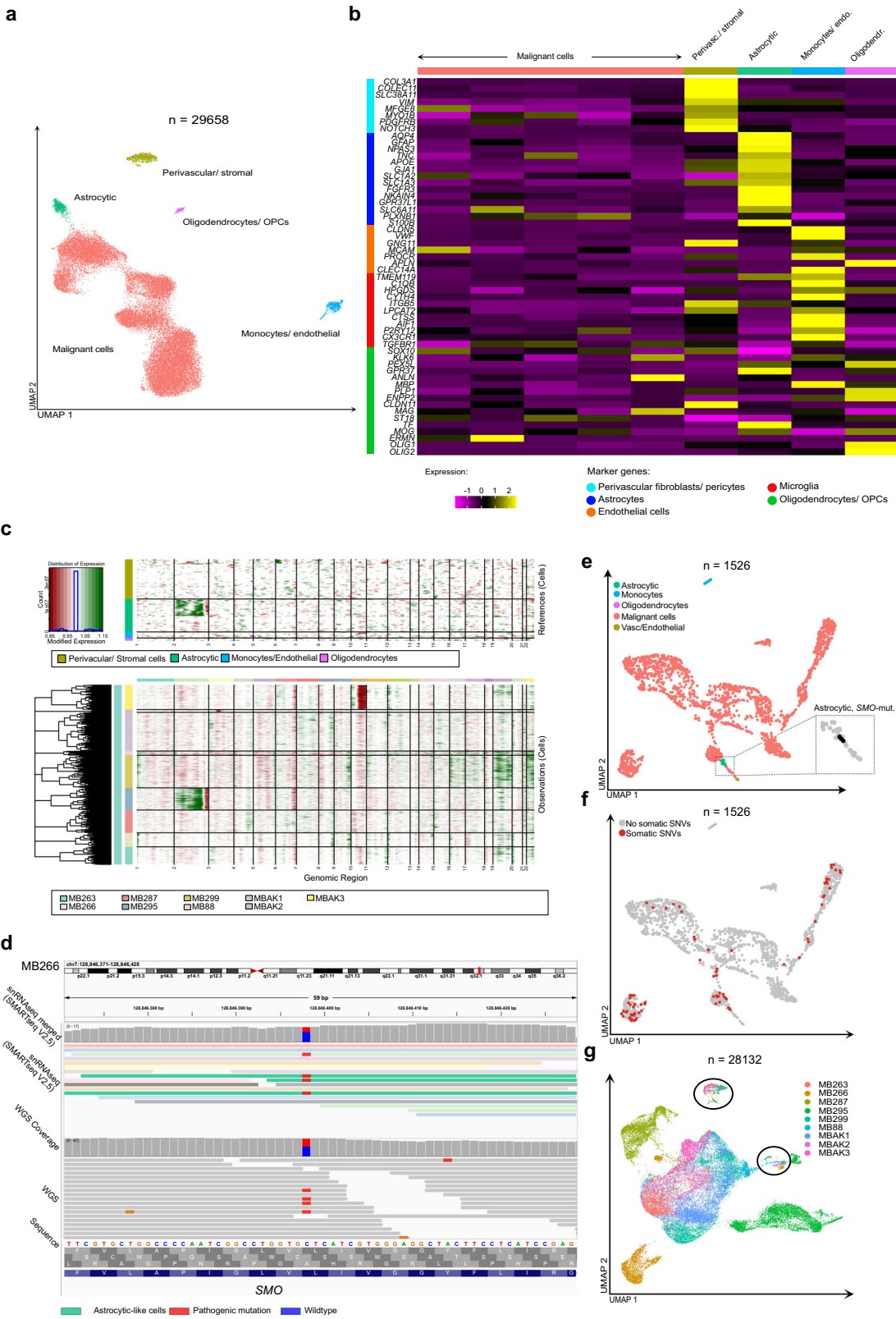

Correlating our snRNA-seq data with transcriptomic profiles of the aforementioned in vivo study[38], MBEN-derived astrocytic-like cells showed clear similarity to both normal and astrocytic-like cells from the respective SHH-MB mouse model (Fig. 3e). In order to further validate our findings, we mapped our MBEN dataset onto a recently published, comprehensive snRNA-seq atlas, spanning the whole embryonal and fetal development of the cerebellum[51]. This analysis

fully reflected cells-of-origin associations identified for SHH medulloblastoma[34] and confirmed results of the pseudotime trajectory analysis. Whereas cells from the beginning of the MBEN trajectory resembled transcriptional signatures of early CGNPs, the intermediate MBEN cluster showed the highest concordance with intermediate developmental stages of CGNP differentiation (Fig. 3f). Lastly, the neuronal-like cluster at the end of the trajectory clearly mapped onto

**Fig. 2 | SnRNA-seq reveals a subset of MBEN-cells with similarity to astrocytes.**
**a** UMAP-projection of the integrated dataset (10X snRNA-seq + SMARTseq V2.5 snRNA-seq) shows intratumoral heterogeneity in MBEN (nine samples, $n = 29,658$ cells). **b** Heatmap depicting marker genes for indicated cell types of all nine clusters. The first five clusters represent malignant MBEN cells. Clusters 6–9 are identified as perivascular fibroblasts/ pericytes, astrocytic cells, oligodendrocytes/ oligodendroglial precursors and a mixture of microglia and endothelial cells. **c** InferCNV-analysis using non-malignant cells as a control confirms malignant origin for the vast majority of cells as well as a CNV signature of malignant cells in the astrocytic cluster. **d** IGV-representation of case MB266 showing pathogenic SNVs within the gene SMO. Three cells with an astrocytic phenotype harbour the same SNV as detected using bulk sequencing. **e** UMAP projection of the SMARTseq

V2.5-dataset without correction for patient batch effects (six samples, $n = 1526$ cells). Cells were mapped to a reference dataset (Sepp et al., bioRxiv 2021) to identify cells with an astrocytic phenotype. As shown, these clustered close to malignant cells. The zoom-in highlights three cells with an astrocytic phenotype and *SMO*-mutations. **f** UMAP projection as in d (six samples, $n = 1526$ cells). Cells in which the same SNVs as in bulk sequencing were detected are highlighted in red. As shown, three cells within the astrocytic population harbor SNVs. **g** UMAP projection of the 10X snRNA-seq dataset which was not corrected for patient-associated batch effects (nine samples, $n = 28,132$ cells). Whereas the majority of cells are clustering according to patient, significant mixing of cells occurs in two clusters (encircled), representing non-malignant cells. Source data are provided as a Source Data file.

differentiated granular neurons and astrocytic-like cells showed high similarity with astrocytes from the developing cerebellum. There was no significant association between the percentage of each cluster per patient and either age of onset, sex or relapse status as calculated using Spearman correlations.

In summary, different MBEN cell populations mimicked distinct cell stages of the normal CGNP lineage, spanning the whole developmental trajectory. The cell stages of MBEN development differed markedly in their major expression programs, and proliferative activity was lost after the early CGNP-like MBEN cells started to differentiate.

## Functional in silico analyses reveal biological mechanisms underlying MBEN differentiation

A number of core biological mechanisms and cellular functions play a pivotal role in CGNP development. These include, amongst others, the SHH pathway, NMDA/glutamate signaling, the expression of bone morphogenic proteins (BMPs) as well as $Ca^{2+}$-signaling[41,42,47,48,52–57]. We sought to investigate whether similar processes might influence MBEN development. First, we examined intra- and inter-cluster communication by analyzing ligand-receptor pairs based on snRNA-seq expression data (Fig. 4a, b)[58]. Astrocytic-like cells were the only cells that showed high-confidence interactions with all other clusters via *APOE* signaling, which is in line with previous findings that astrocytic cells act as the main supplier and redistributor of cholesterol in the CNS[59]. Intracluster communication within the differentiated neuronal-like cluster was dominated by genes which are involved in voltage-dependent signaling, especially with regard to $Ca^{2+}$-signaling, such as *RIMS1, CACNA1C* and *CALM1*[60]. These findings were corroborated by analyzing the expression of a transcriptomic $Ca^{2+}$-signaling signature including 1805 genes[61]. Indeed, $Ca^{2+}$-signaling was strongly connected to later cell stages within MBEN (Fig. 4c). Another group of genes that is involved in CGNP development relates to BMPs that may also suppress MB proliferation in vitro[41,62]. Interestingly, a BMP-related signature showed strong expression in astrocytic-like cells, suggesting that these cells might be involved in triggering the differentiation process by suppressing proliferative activity in MBEN (Fig. 4d). The hypothesis that similar processes as in physiological CGNP development are active in MBEN was further confirmed by cluster-specific gene ontology (GO) analysis (Supplementary Fig. 6, Supplementary Data 3)[63]. All clusters showed significant enrichments for GO terms directly connected to synaptic organization and activation, glutamate signaling, and $Ca^{2+}$-homeostasis (Supplementary Fig. 6, Supplementary Data 3). Astrocytic-like cells showed high expression of genes connected to cAMP-metabolism, possibly indicating that malignant astrocytes may be involved in energy supply for surrounding cells, thus resembling astrocytic functions in normal CNS[64].

In order to complement these findings, we performed a cluster specific transcription factor (TF) activity analysis using the DoRothEA tool[65,66]. To this end, overall TF activity per cluster and changes in TF activity between MBEN stages that followed each other were calculated (Fig. 4e, Supplementary Fig. 7, Supplementary Tables 4, 5). *SOX2* and *FOXA1*, two pioneer TFs which are involved in maintaining chromatin

accessibility throughout early development[67], were active along the entire differentiation process. In contrast, *E2F1* and *E2F4*, which are particularly involved in cell cycle regulation[68], were downregulated shortly after the start of the trajectory. In addition, *MYC*, which was among the most active TFs in (proliferating) early-like CGNPs, was downregulated in later MBEN cell stages. *EOMES*, a marker of unipolar brush cells and intermediate cortical neurons[34,69], increased in activity once (proliferating) early CGNP-like cells differentiated into migrating CGNP-like cells and astrocytic-like cells. Similarly, *NEUROD1*, a TF that has been described as an inducer of differentiation and thus tumor suppressor gene in MB, increased activity throughout the MBEN differentiation process (Fig. 4e, Supplementary Data 4, 5). Surprisingly, despite clear differences in expression profiles, astrocytic-like cells showed overall similar TF activities to other clusters, possibly due to their shared developmental roots. Notably, *NR1H3*, a TF that is strongly involved in cholesterol homeostasis[70], showed markedly increased activity in astrocytic-like cells.

Taken together, complimentary functional in silico analyses confirmed that biological mechanisms of physiological CGNP development are mimicked in MBEN differentiation. Furthermore, astrocytic-like cells showed upregulation of BMPs and were connected to energy and cholesterol homeostasis.

## Transcriptomic analysis of MBEN based on microdissected tissue shows differences between the internodular- and nodular compartments

In order to correlate our findings with the bicompartmental histology of MBEN, we applied targeted microdissection to FFPE samples overlapping with our study cohort, allowing to obtain bulk RNA-seq datasets from nodular and internodular tumor areas separately ($n = 26$). After expression variance inspection and fingerprint verification of corresponding samples (Supplementary Fig. 8a, b) we corrected for patient-related batch effects of the original tissue (details in methods) and detected genes differentially expressed between nodular and internodular compartments (Fig. 5a, Supplementary Data 6). When we investigated the expression of these genes in our snRNA-seq datasets, we observed clear patterns. For example, *TRIM9*, which was nodular-specific in the microdissection data, was strongly expressed in intermediate CGNP-like and neuronal-like snRNA-seq clusters (Fig. 5b). Similarly, *TMEM108* from internodular derived microdissections demonstrated a strong fit to clusters of early CGNP-like MBEN cells and astrocytic-like cells (Fig. 5c). To further confirm these findings, we performed gene set variation analysis[71] that demonstrated differentiated MBEN clusters being strongly enriched with nodular, whereas the early CGNP-like populations were in close match to internodular transcriptome profiles (Fig. 5d). Interestingly, astrocytic-like cells were more specific to the internodular compartment. These findings were corroborated by performing a deconvolution analysis on our microdissected bulk-RNA sequencing datasets with the cluster signatures of our snRNA-seq dataset as reference in order to estimate the contributions of differentiated, neuronal-like and proliferating, early CGNP-like cells to the two histological components[72]. We found

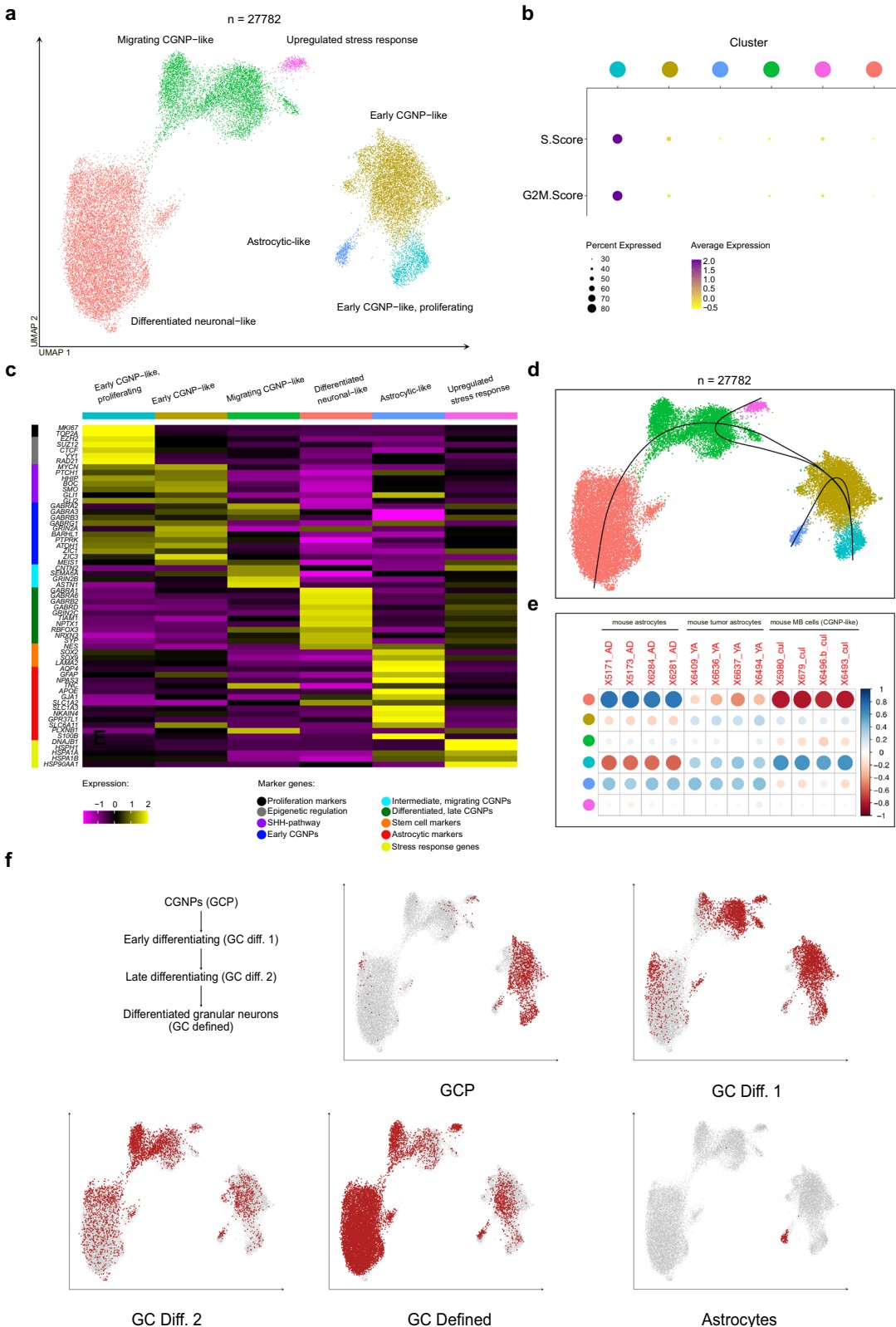

significant differences that reflected the results of our microdissection-based sequencing analysis (Supplementary Fig. 8c, d).

## Spatial transcriptomics correlates cell stages in MBEN to its histologic compartments

Our microdissection-based expression analysis revealed clear differences between the two MBEN compartments, however, it did not allow

to investigate the detailed spatial distribution of the MBEN cell stages that were identified using snRNA-seq. Thus, we performed spatial transcriptomic analysis via single molecule RNA-in situ hybridization on three representative samples (smRNA-FISH/RNAscope). For RNA-scope, twelve genes were chosen as markers for the different snRNA-seq clusters, and also considering biological significance (e. g., *GLI1* and *PTCH1* as SHH-pathway members) (Fig. 6a, b, Supplementary

**Fig. 3 | MBEN differentiates along a cerebellar developmental trajectory.**
**a** UMAP-plot of the merged snRNA-seq-dataset (10X snRNA-seq + SMARTseq V2.5 snRNA-seq) restricted to malignant cells only (*n* = 9 patients). Clusters are named based on their similarity to major stages of physiological cerebellar granular neuronal precursors (CGNP) development. **b** Dotplot illustrating proliferation activity based on gene signatures of S- and G2M-cell cycle activity, which is restricted to one cluster only. Color code of the x-axis as in **a**. **c** Heatmap showing the expression of important cerebellar developmental genes in the dataset per indicated cluster. **d** UMAP-projection with overlayed pseudotime trajectories

shows two major lineages, with the majority of cells following a granular cerebellar trajectory, and a small subset differentiating into astrocytic-like cells. **e** Astrocytic-like cells show transcriptional similarity to murine astrocytes and murine astrocytic-like cells, but not to MB cells with transcriptional similarity to CGNPs (Yao et al., Cell, 2020). Color code of the y-axis as in **a**. **f** Feature plots of the dataset highlighting similarity of MBEN associated tumor cells to non-malignant cells of the developing cerebellum. GCP = granular cerebellar precursors (Sepp et al., bioRxiv, 2021) (*n* = 9 patients/27,782 cells each). Source data are provided as a Source Data file.

Data 7). Two genes were excluded due to patient-specific expression or lack of discriminatory power in the spatial dataset, with 10/12 genes being selected for downstream analysis (Supplementary Figs. 9, 10a–j). After spot detection and nuclei segmentation, transcripts per nucleus were quantified and used to construct spatially derived single cells, which were then mapped back to the scanned image (Supplementary Figs. 9, 10a–j). Based on marker gene expression alone, the bicompartmental histology of MBEN could be readily reconstructed. Using *RBFOX3*, a gene that encodes for the neuronal marker protein NeuN being specific for differentiated neuronal cells, and the embryonal gene *LAMA2*, found to be differentially expressed in non-malignant stromal and astrocytic(-like) cells, we could distinguish between the nodular (*RBFOX3* high, *LAMA2* low), and internodular (*RBFOX3* low, *LAMA2* high) compartment (Supplementary Fig. 11a–f). These findings were further confirmed when we used the full set of all ten genes. Clusters of the RNAscope-derived single cells fully reconstructed both the snRNA-seq and microdissected tissue-derived findings, i.e., pre-migratory cycling and non-cycling early CGNP-like clusters, *LAMA2*-positive cells that included both stromal and astrocytic cells and migrating as well as postmigratory neuronally differentiated cells (Fig. 6c). Whereas the first three cell types were restricted to the internodular tumor compartment, postmigratory, differentiated cells were only found in the nodular areas (Fig. 6d–h). Intermediate, migrating cells were observed in both compartments. These visual observations were supported by quantifying the probability of each of the five cell types to be located next to each other, respectively, which confirmed that non-proliferating and proliferating early CGNP-like and *LAMA2*-positive cells (including astrocytic, vascular, and stromal cells) were co-localizing (Fig. 6g). In contrast, early CGNP-like and late stage neuronally differentiated MBEN cells were clearly less likely to be located next to each other (Fig. 6g).

Taken together, a targeted spatial transcriptomics approach confirmed our observations based on snRNA-seq and revealed that different normal CGNP-related cell stages of the MBEN developmental lineage could be mapped to the internodular and nodular compartments, respectively.

### The tumor microenvironment in MBEN differs between internodular and nodular compartments

The restricted number of marker genes using smRNA-FISH did not allow to differentiate between astrocytic and stromal cells. We therefore extended the spatial analysis by using the Molecular Cartography platform from Resolve Biosciences on four representative samples (Supplementary Data 8). In line with our prior findings, the expression patterns of single genes, such as *LAMA2* and *NRXN3*, were already sufficient to reconstruct the bicompartmental MBEN histology on the transcriptomic level (Fig. 7a). In total, 92,666 DAPI-segmented cells were subjected to downstream analysis. We were able to recover all major MBEN cell stages, namely proliferating and non-proliferating early CGNP-like, migrating CGNP-like and neuronally differentiated tumor cells (Fig. 7b, c, Supplementary Fig. 12a–e, Supplementary Fig. 13a–e, Supplementary Data 9). Furthermore, one small cluster was dominated by proliferative activity only, whereas another cluster expressed genes of CGNPs in later developmental stages and was thus

termed "late CGNP-like". These two clusters, which were not distinguished as clearly in our snRNA-seq data, most likely represented subsets of proliferating, early CGNP-like and late stage neuronally differentiated MBEN cells. Notably, the gene *TMEM108* (identified as differentially expressed in our microdissection-based gene enrichment analysis) could be confirmed as a marker for the internodular MBEN compartment (Supplementary Fig. 13a). In addition, the designed extended gene panel allowed for identification of non-malignant cell types, such as stromal, vascular/endothelial, immune cells and astrocytic cells (Fig. 7b, c, d). In general, the molecular cartography workflow retrieved larger number of non-malignant cells from the tumor microenvironment, than snRNA-seq (Supplementary Fig. 12c–e, Supplementary Fig. 14a). This was particularly relevant for stromal, endothelial, and immune cells, of which the majority were identified as monocytes. Interestingly, the number of astrocytic cells (non-malignant astrocytes and astrocytic-like tumor cells) was comparable across both platforms (Supplementary Fig. 12d,e). The monocytic cells could be further subdivided into two distinct populations, which were primarily distinguished by the expression of *CD16* and *CD163* (Supplementary Fig. 14c–k). Whereas CD16 has been described as a marker of proinflammatory monocytes leaning towards the M1 polarization phenotype, CD163 is generally seen as a marker of anti-inflammatory M2 monocytes[73]. Overall, the ration of *CD163*+ to *CD16*+-monocytes was 9:1, hinting at a generally more anti-inflammatory tumor microenvironment (Supplementary Fig. 14b). Next, we aimed at determining whether these two subpopulations could be identified as microglia or bone marrow-derived macrophages. To this end, we performed an enrichment analysis of MSiGdb cell type marker genes[74] with subsequent statistical testing using chi-squared tests. Based on the statistically most significant terms, *CD163*+ cells mostly resembled microglia ($p = 4.88 \times 10^{-14}$). Furthermore, across this population, weak expression of the microglia marker *TMEM119* was observable (Supplementary Fig. 14e). The most significant hits for the *CD16*+ population were associated with dendritic cells ($p = 2.01 \times 10^{-14}$) and macrophages ($p = 4.79 \times 10^{-14}$) (Supplementary Data 10). However, due to the comparably low number of genes and the limitations of gene enrichment analysis, these results remained inconclusive. Oligodendrocytes seemed to be generally underrepresented in the MBEN tumor microenvironment. Non-malignant astrocytes could not be distinguished from astrocytic-like cells due to missing CNV information. However, we observed that a fraction of tumor cells was transcriptionally similar to astrocytic cells (Fig. 7d). Notably, astrocyte marker genes, such as *NPAS3*, *CD44* and *LAMA2*, were also expressed in a fraction of early CGNP-like cells, which were identified as potential precursors of astrocytic-like cells in our snRNA-seq trajectory analysis (Figs. 3d, 7b, e, Supplementary Fig. 13b–d).

Next, we sought to investigate the spatial distribution of each cell type based on the annotation described above. Similar to the results that we observed using smRNA-FISH, the internodular compartment was formed by (proliferating) early GCNP-like cells, whereas the nodular compartment consisted of neuronally differentiated tumor cells. Intermediate cell stages that expressed markers of migrating CGNPs were found in both histological areas (Fig. 7f–i). We then used cell proximity network analysis to investigate which cell types co-localize

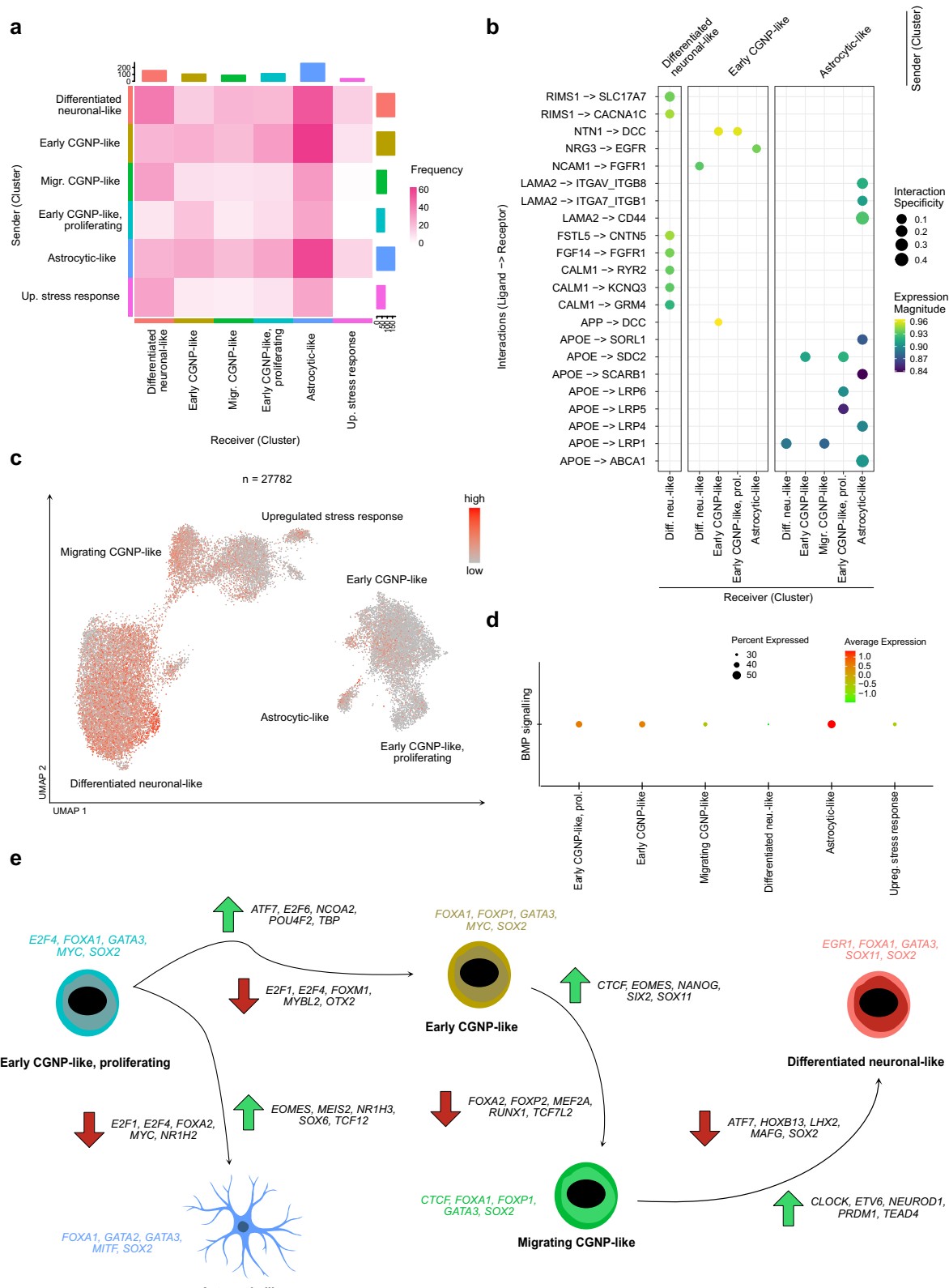

in MBEN (Supplementary Fig. 13e). Strikingly, there were marked differences between astrocytes/astrocytic-like cells and other non-malignant cells in terms of spatial distribution (Supplementary Fig. 13e, Supplementary Fig. 15a–j, Supplementary Fig. 16a–j). Stromal cells, vascular cells, and monocytic cells, which clustered separately from tumor cells in the UMAP-projection, were strongly enriched in the internodular compartment and co-localized with early CGNP-like,

proliferating MBEN cells. In contrast, cells with an astrocytic phenotype were found in close proximity to migrating, late CGNP-like and postmitotic neuronally differentiated cells. However, the majority of astrocytic cells did not localize within the nodular compartment. Instead, they were found at the rim of the tumor nodules, forming a transition zone between the two histological MBEN parts (Fig. 7j, k). In order to further validate the localization of astrocytic cells in MBEN, we

**Fig. 4 | Biological processes that regulate MBEN differentiation mimic CGNP development. a** Heatmap depicting the frequencies of predicted interactions for each pair of potentially communicating cell stages amongst MBEN cells. **b** Dotplots depicting the most confident receptor-ligand-interactions within and between different MBEN cell stages. The interaction specificity weights indicate how specific a given interaction is to the sender and receiver cell stages. **c** UMAP-plot showing the expression of a Ca²⁺ signaling signature along MBEN differentiation (nine samples, $n = 27,782$ cells). To improve visibility, only cells with expression levels above the 25th quantile are highlighted. **d** Dotplot showing the expression of a BMP protein signature along MBEN differentiation. **e** Visualization of transcription factor (TF) activities within and between MBEN cell stages. Above each cell stages, the top five expressed TFs are indicated. Furthermore, the five most up- and down-regulated TFs that changed during each developmental step of the MBEN trajectory are given (see also Fig. 3d). In order to design **e**, the image "Blue Astrocyte" (by Andrew Hardaway, licensed under https://creativecommons.org/licenses/by/4.0/) from the database https://scidraw.io/ was used. Source data are provided as a Source Data file.

stained a cohort of MBENs with available FFPE-derived tissue ($n = 12$) astrocytic marker GFAP. In full concordance with our spatial transcriptomics data, GFAP was expressed in the peripheral zone of the nodular MBEN compartment (Supplementary Fig. 17a–d). GFAP-positive cells showed various appearances, ranging from mature astrocytes with numerous processes to small, primitive-like cells. Interestingly, reactive astrocytes as characterized by Nestin-expression[75], were only found in 1/5 cases of the TCL1-, but 6/7 cases of the TCL2-subtype (Supplementary Fig. 17a–d, Supplementary Table 1). To follow this up, we analyzed the expression of *GFAP* and *Nestin* in MBENs as compared to DNMBs. Whereas these markers were significantly correlated in MBEN, this was not the case in DNMBs, indicating that there are differences between the astrocytic cell population in these related, but different tumors (Supplementary Fig. 17e, f).

Taken together, expanded spatial transcriptomic analysis revealed that the tumor microenvironment in MBEN differs between areas of early CGNP-like and neuronally differentiated tumor cells (Fig. 8). This data suggests that cells that contribute to the tumor microenvironment in MBEN influence different stages of its differentiation process.

## Discussion

In this study, we applied an integrated multi-modal transcriptomics approach to unravel the underlying biology of the histomolecular heterogeneity of MBEN. Our data indicate that histologically apparent nodular and internodular areas reflect a spectrum of cell stages that are connected through an underlying developmental trajectory which mimics the physiological differentiation of the CGNP-lineage during cerebellar development. Throughout this process, MBEN cells lose proliferative activity and differentiate into a neuronal-like phenotype, which is thought to explain the favorable prognosis in most patients including case reports of maturation into benign gangliocytomas[23,24]. Our results are supported by an accompanying study from Gold et al., who identified the same MBEN lineage in an independent snRNA-seq dataset and used spatial proteomics assays to observe similar patterns of cellular organization as in our spatial transcriptomics experiments. These insights are of high interest given that a distinct biological property of pediatric cancer is a block of developmental maturation rather than gained ability to de-differentiate. In line with this conclusion, (epi-)genetic maturation blocks have been identified as an emerging topic in pediatric oncology representing potential targets for new therapeutic avenues[76]. Our data suggest that reactivation of CGNP-associated signaling pathways mimicking normal development and distinct intercellular communication processes result in this phenomenon. It remains enigmatic though, why these processes differ in more aggressive variants of SHH-MB, such as *TP53*-mutated SHH-MB. Interestingly, Gold et al. find evidence that late stage, neuronally differentiated MB cells are more common in MBEN than in more aggressive SHH-MB, indicating a potential maturation block in the developmental lineage of these cells. One hypothesis is that tumor formation in MBEN largely depends on SHH-pathway activation in proliferating CGNPs, but without the potential to form a more aggressive phenotype[15]. The consistent upregulation of SHH-mediated transcription factors may be strong enough to induce neoplastic transformation in early CGNPs in the EGL, which exhibit high proliferative and migratory potential even under physiological circumstances. Similar to early CGNPs, undifferentiated MBEN cells exhibited upregulation of SHH-signaling, whereas late stage MBEN cells showed activation of cellular processes such as Ca²⁺- and NMDA receptor-signaling, which are characteristic for more differentiated CGNPs in the developing cerebellum[77]. These findings indicate that shared molecular functions between embryogenesis and MBEN formation may exist. Furthermore, it seems likely that epigenetic regulation, for instance via the PRC2-complex, may play a role in MBEN-differentiation, which is underlined by the fact that *EZH2* was upregulated in proliferating, early CGNP-like MBEN cells[42,45].

Several studies recently applied single cell RNA-sequencing to MB[27,28,39,78]. Hovestadt et al. reported that infant SHH-MB showed transcriptional similarity to intermediate and late stage CGNPs, while adult tumors correlated with undifferentiated CGNPs and unipolar brush cell-progenitors[27]. Interestingly, studies by Hovestadt et al. and Vladoiu et al., which focused mainly on group 3/4-MB, did not report on the existence of tumor cells with astroglia signatures, while this cell type was also identified in a study on murine SHH-MB models by Ocasio et al.[39]. This raises the intriguing hypothesis that astrocytic-like cells associated with MB may be characteristic for SHH-MB. One histopathologic study reported that astrocytic differentiation and GFAP-positivity were restricted to MB cells from nodular tumors and not present in classic MB, however, this study dates back to a time when MBEN was not yet established as a distinct entity[79]. The hypothesis of CGNP-derived tumor cells developing into astrocytic-like cells is supported by the fact that CGNPs can differentiate into astroglial cells upon exposure to elevated levels of SHH[80]. Whereas the main developmental, CGNP-like trajectory was validated across two methods and datasets, the exact position of astrocytic-like cells within this trajectory was more ambiguous. Since these cells did not show proliferative activity, were located within the bicompartmental MBEN transition zone and based on the findings by Yao et al., who showed in a murine SHH-MB model that astrocytic-like tumor cells were not stem cells, it seems likely that this cell population does not represent a cell of origin for MBEN[38]. However, additional studies—including functional modeling—will be necessary to determine the exact developmental role of this enigmatic cell type. Based on our current knowledge, it seems likely that malignant astrocytic cells stem from early CGNPs in the EGL[34,38,80]. The original publication that coined the term "MBEN" by Giangaspero et al. already described astrocytic, GFAP-positive tumor cells in the internodular compartment, which is why we suggest giving MBEN-associated astrocytic-like cells the byname "Giangaspero cells"[16].

Microglia, astrocytes, and astrocytic-like malignant cells play an important role in the formation and progression of SHH-MB[38,40,81–83]. We attempted to distinguish between malignant and non-malignant astrocytes based on their clustering behavior and CNVs. This approach was without alternatives in light of the current lack of MBEN models, and future investigations will be needed to further unravel the functional and genomic differences between these two cell types in the human setting. However, several studies have generated convincing evidence that both normal and tumor-derived astrocytes may support

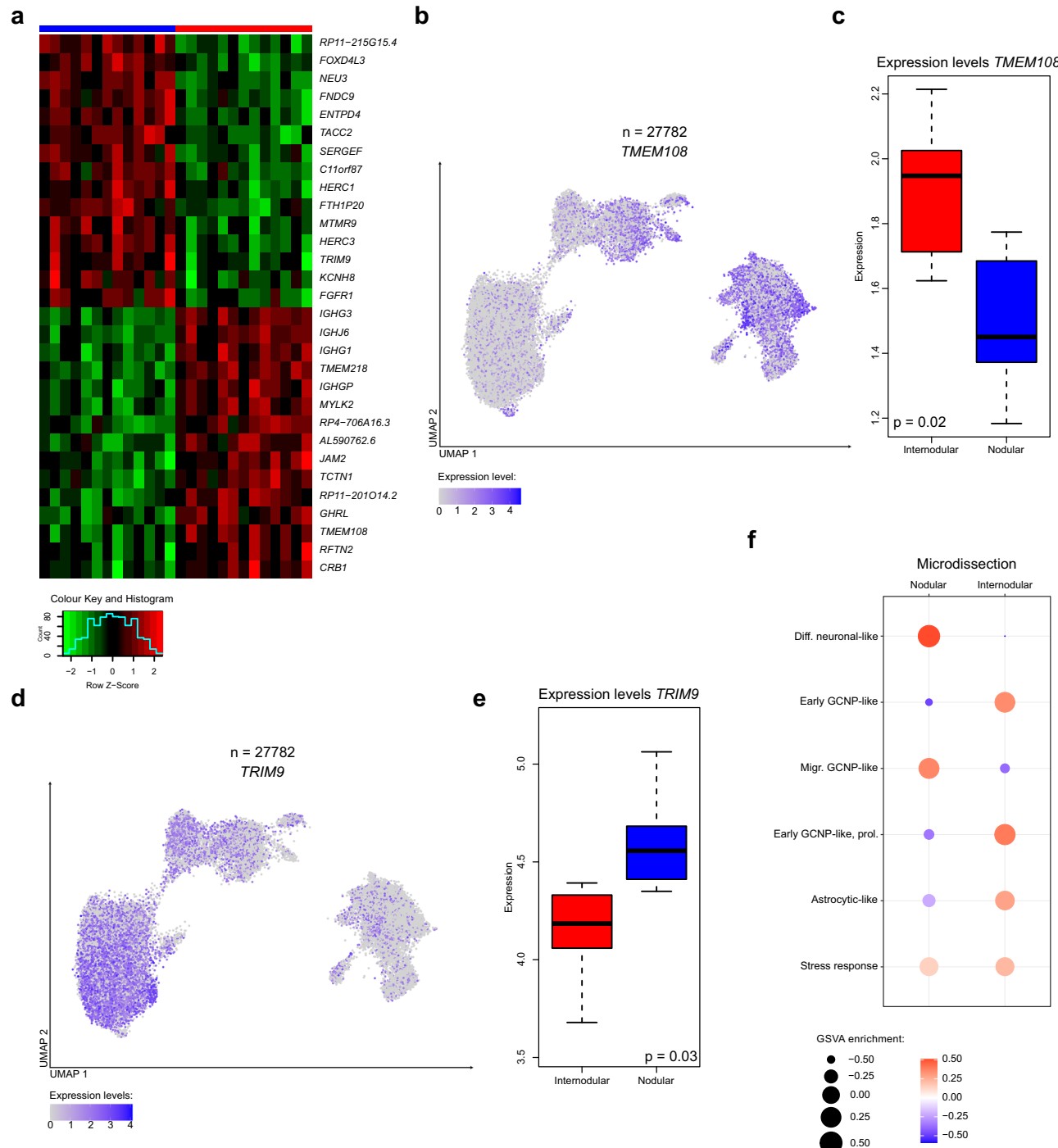

**Fig. 5 | Microdissection reveals transcriptional differences between internodular and nodular tumor areas that correspond to distinct MBEN cell stages.** **a** Heatmap showing differential gene expression between internodular (red column) and nodular (blue column) MBEN compartments. **b**, **c** *TMEM108*, a marker of the internodular compartment, is mainly expressed in early CGNP-like cells. Feature plot on the left depicting gene expression in the snRNA-seq dataset. Boxplot on the right showing expression (measured in batch-effect adjusted RPKM) in micro-dissected MBEN tissue (limma statistical method *p*-val = 0.02 with adjustment for multiple genes and tumor samples batch effect) inspected in *n* = 13 biologically independent samples. **d**, **e** *TRIM9*, a marker of the nodular compartment, is mainly expressed in later stages of MBEN development. Feature plot on the left depicting

gene expression in the snRNA-seq dataset. Box plot on the right showing expression in microdissected MBEN tissue (limma statistical method *p*-val: 0.03 with adjustment for multiple genes and tumor samples batch effect) inspected in *n* = 13 biologically independent samples. **f** Mapping of the expression signatures of microdissected internodular and nodular histological compartments to the different snRNA-seq clusters via Gene set variance analysis (GSVA) shows distinct transcriptional similarities between differentiated cells and the nodular compartment, and early-CGNP like cells with the internodular areas. **b**, **c** The center line, box limits, whiskers, and points indicate the median, upper/lower quartiles, 1.5× inter-quartile range and outliers, respectively. Source data are provided as a Source Data file.

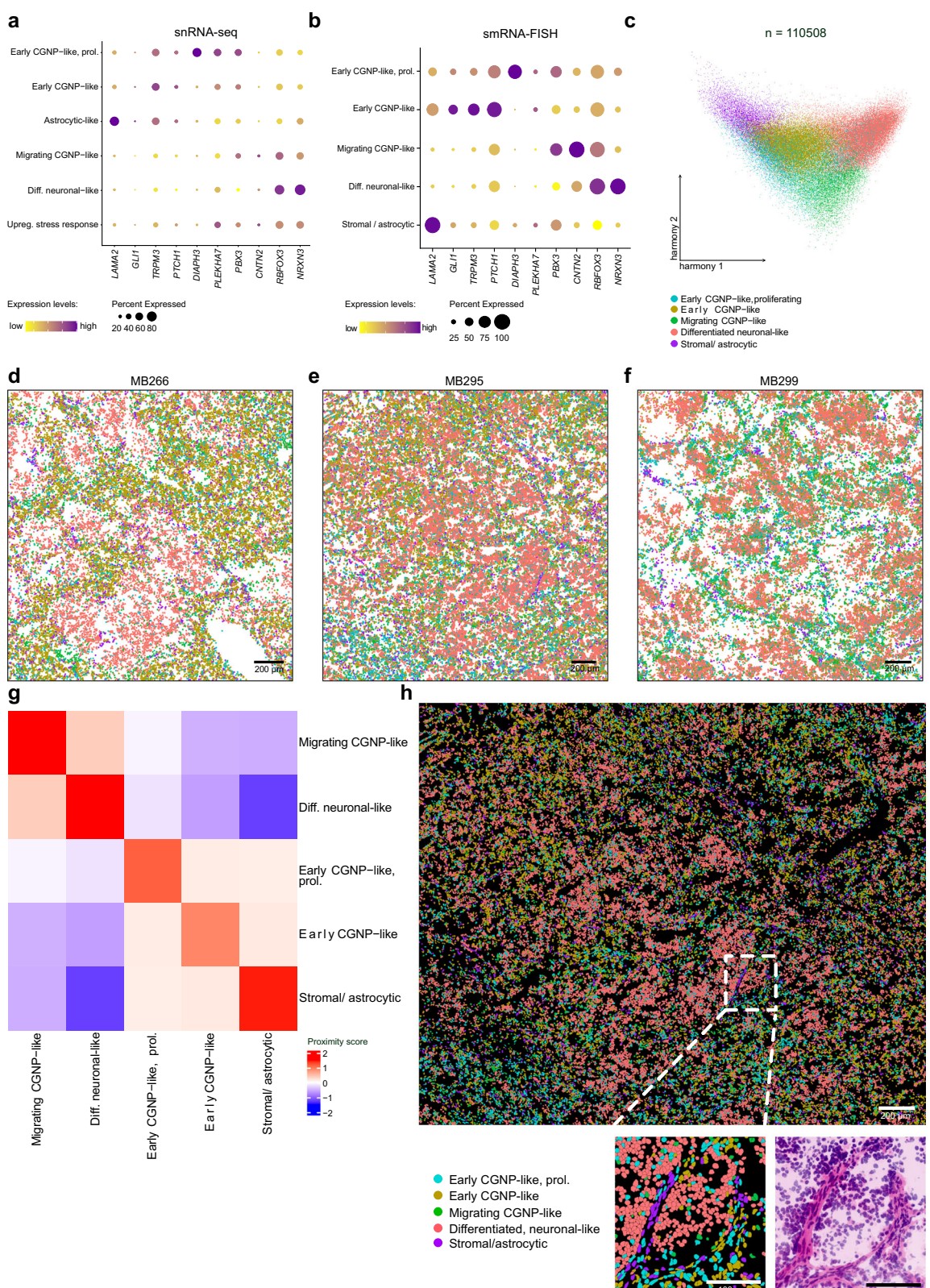

SHH-MB proliferation and growth in murine models[38,81–83]. Our data indicate that cells with an astrocytic phenotype are found at the border between the two compartments, co-localizing with MBEN cells in their migrating and later, postmitotic differentiation stages. In contrast, all other non-malignant cell types were in close proximity to early CGNP-like tumor cells. Given the fact that astrocytes are known to influence surrounding cells via paracrine signaling, these patterns suggest that astrocytic cells may have a direct influence on the differentiation process of MBEN, whereas other non-malignant cell types are mostly involved in supporting its proliferating subpopulations, or, in the case of microglia, trying to suppress tumor growth[84]. Whereas it was not possible to safely distinguish between different subtypes of monocytic cells, we identified two distinct populations that resembled the pro- and anti-inflammatory M1 and M2 phenotype, respectively.

**Fig. 6 | Spatial transcriptomics map developmentally distinct cell stages to spatially distinct tumor compartments.** Dotplots showing the expression of the ten genes chosen for spatial transcriptomics in the snRNAseq- (**a**) and the smRNA-FISH-dataset (**b**) for each cluster, respectively. **c** Harmony corrected UMAP-clustering of smRNA-FISH-derived single cells shows distinct clusters (three samples, n = 110,508 cells). **d**–**f** Mapping single cells back to the original scans reveals spatially distinct localization of cells from each cluster with regard to the bicompartmental structure of MBEN. Color code corresponds to **c**. Each image represents one sample. **g** Network visualization quantifying the probability of co-localization of cells from each cluster. Red and blue colors indicate high and low probability for co-localization, respectively. (Proliferating) early CGNP-like and stromal/astrocytic cells are co-localizing, whereas early and late stages of the MBEN trajectory are spatially separated. **h** A scan of the sample MB295 shows clear correlation of the smRNA-FISH-derived clusters to spatially distinct regions of MBEN, as they were found in all three investigated samples. Lower panel left: Zoom-in with pseudo H.E. staining. Lower panel right: Corresponding mapping of single cells illustrates the spatial architecture of the transition zone between the two histological compartments of MBEN, which is mainly formed by early CGNP-like, migrating and stromal/astrocytic cells. Scale bars in **d** –**f** =200 μm. Scale bars in **h** =200 μm (large image), 100 μm (Zoom-ins). Source data are provided as a Source Data file.

Interestingly, the number of anti-inflammatory monocytes was significantly higher. With these observations in mind, the internodular compartment of MBEN can be understood as a source of neoplastic cells, which then migrate into the nodular compartment while losing proliferative potential.

DNMB, another histological variant of SHH-MB, also shows a bicompartmental structure, but represents a biologically and clinically distinct phenotype. Results of Gold et al. indicate that the nodular areas may vary in their level of differentiation, with MBEN tumors being more likely to contain cells resembling later stages of differentiation. Interestingly, we observed that the astrocytic markers GFAP and Nestin were expressed in correlation in MBEN, but not in DNMB. These findings indicate that there may be differences between MBEN and DNMB with regard to the role of astrocytic cells within the tumor microenvironment. To date, murine or cell models that faithfully preserve the unique histological composition of MBEN and could be used to perform functional analyses are lacking.

By comparing snRNA-seq and spatial transcriptomics methods, we found that the number and types of cells from the tumor microenvironment that could be identified with each protocol differed. This was particularly true for monocytes, highlighting the need to integrate a spectrum of methods to investigate their role in neuro-oncological diseases.

Our study resolves the intratumoral heterogeneity of MBEN at the single cell level and reveals the spatial relation between the different cell types in the context of its bicompartmental histological structure. Thus, it provides a framework for similar analyses in other malignancies with intratumoral heterogeneity[85]. Finally, further deepening our understanding of the biological principles underlying both intratumoral differentiation processes and maturation blocks are expected to guide the development of drugs that either induce or overcome these phenomena in embryonal cancer types.

## Methods
### Material and data collection, inclusion, and ethics
This study was performed after approval by the ethics committee of the Medical Faculty of Heidelberg University. All experiments in this study involving human tissue or data were conducted in accordance with the Declaration of Helsinki and relevant national and international ethical regulations. Cases from earlier studies on MBEN were screened regarding the availability of fresh frozen tissue[15,25,86]. Clinical data and tissue of all nine cases in the study were collected from patients from the international DKFZ cohort after receiving written informed consent, including the publishing of clinical and epidemiological data, from the respective patients or their legal representatives and after approval by the ethic board of the Medical Faculty of Heidelberg University. Clinical and epidemiological data, including information on sex/gender, was retrospectively derived from clinical records. DNA-methylation profiling, DNA-panel sequencing and bulk-RNA sequencing data was partly derived from earlier studies[12,86].

### DNA-methylation profiling and CNV analysis
DNA methylation profiling was performed using the Infinium Human Methylation 450k and EPIC BeadChips as previously described[9].

Subsequently, the Heidelberg Brain Tumor Methylation Classifier v11b6 (https://www.molecularneuropathology.org) was applied for molecular classification. Copy-number variation analysis from 450k and EPIC methylation array data was performed using the conumee Bioconductor package version 1.12.0 (Hovestadt V, Zapatka M, 2017).

### RNA-bulk sequencing
Bulk RNA-seq data were analyzed as described previously[86]. Shortly, reads alignment was performed with STAR tool[87] and gene expression counts were computed with the Subread package[88].

### DNA-panel sequencing
Molecular barcode-indexed ligation-based sequencing libraries were constructed using 200 ng of sheared DNA. Libraries were enriched by hybrid capture with custom biotinylated RNA oligo pools covering exons of 130 cancer-associated genes. Paired-end sequencing was performed using the NextSeq 500 (Illumina). Sequence data were mapped to the reference human genome using the Burrows–Wheeler Aligner and were processed using publicly available SAM tools[89]. Only variants annotated as "exonic" or "splicing" were included, "intergenic" and other untranslated regions were excluded. Recurrent gene mutations of *PTCH1*, *SUFU*, and *SMO* were also assessed with residual DNA from the same pool used for sequencing by polymerase chain reaction followed by direct Sanger sequencing of the corresponding exons.

### Nuclei isolation
Prior to nuclei isolation[90], all necessary materials (douncer and pestles, falcon tubes, pipette tips) were pre-coated using coating buffer (sterile phosphate buffered saline (PBS), filtered with Millex HA filter units 0.45 μl). Fresh frozen tissue was placed on dry ice and cut with a scalpel. Next, the tissue was put into 5 ml of lysis buffer containing DTT and Triton-X (0.1%), and subsequently lysed using a glass douncer and pestle. The nuclei were filtered two times (100 μM and 40 μM filters) and centrifuged for 5 min at 500 x g and 4 °C, followed by two washing steps (5 min, 500 x g, 4 °C) using a PBS-containing washing buffer. The resulting pellet was resuspended in 1 ml of storage buffer. Nuclei were checked for integrity and the absence of cell debris under a light microscope and subsequently counted using a Luna Automated Cell Counter (Logos Biosystems).

### SnRNA-seq 10X Genomics
The 10X Genomics 3′-Single Cell RNA-sequencing V2 protocol was applied to all samples according to the manufacturer's instructions using 10X Chromium Single Cell 3′ Reagent Kits v2 (https://www.10xgenomics.com/). 14,000 nuclei were loaded per sample and processed with the Chromium Controller. The resulting cDNA-libraries were quantified with Qubit fluorometric quantification (ThermoFisher Scientific) and quality assessment was done using the TapeStation system (Agilent). Sequencing was performed according to the manufacturer's instructions.

### SnRNA-seq SMARTseq V2.5
Whole transcriptome snRNA-seq was performed following an adapted SMARTseq protocol (SMARTseq V2.5) for nuclei[91]. For each sample,

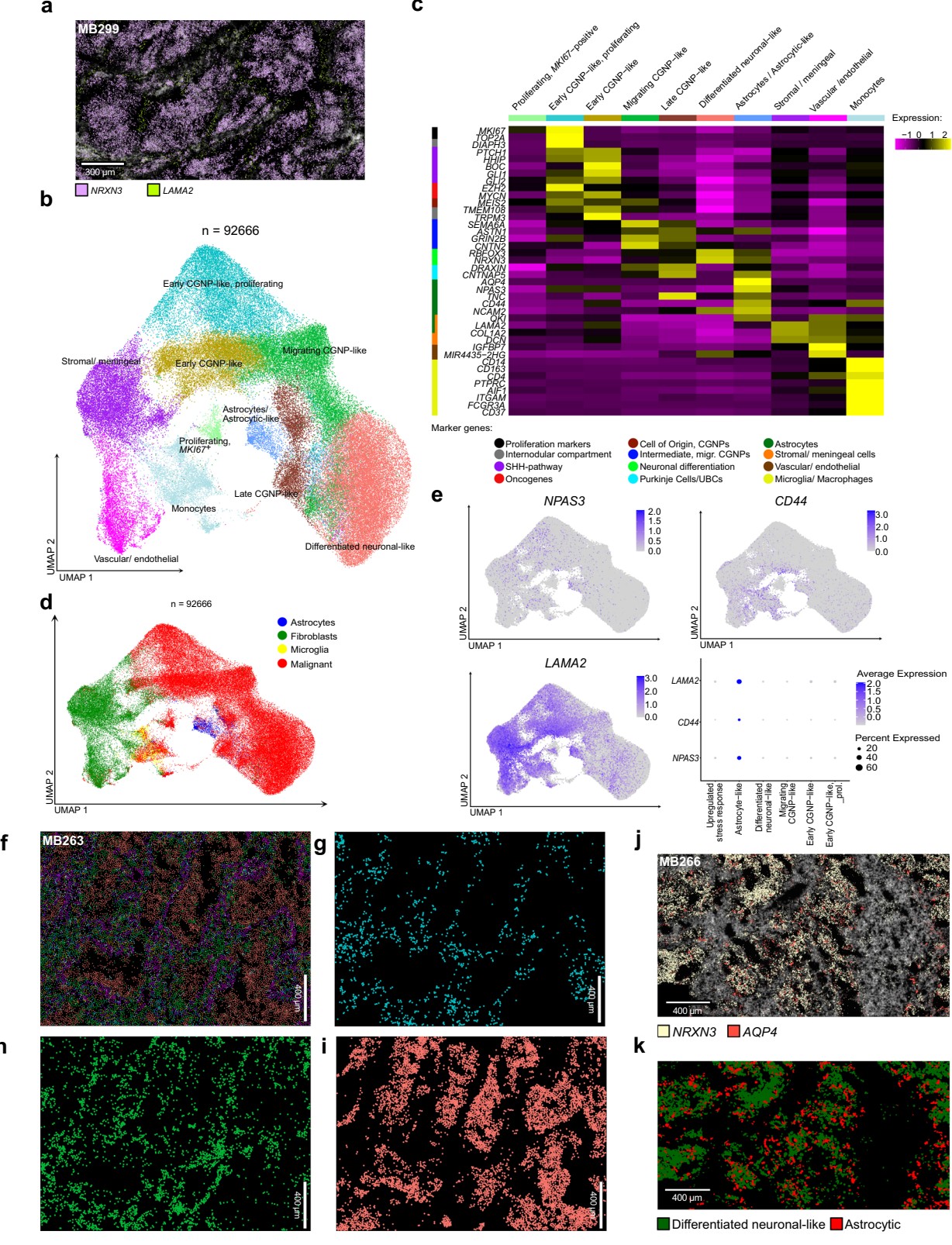

single nuclei were sorted in 384-well plates into 1 μl of lysis buffer containing RNaseinhibitors and a polyT oligo. For the reserve transcription reaction, we used a TSO with LNAs (Eurogentec) at a final concentration of 3 μM and the Maxima H Minus reverse transcriptase (Thermo) under molecular crowding conditions with 7.5% PEG-8000. Subsequently, PCR was performed with the KAPA HiFi HotStart ReadyMix (KAPA Biosystems), followed by a purification step with AMPure XP beads (Beckman) using the Agilent Bravo system. Library preparation was done via the Nextera XP Library Prep kit (Illumina) using 350 ng of cDNA as input. The reaction volume of each step was reduced by a factor of three and all pipetting steps were done with the mosquito LV (SPTlabtech). After pooling all libraries from one plate,

**Fig. 7 | The distribution of non-CGNP-like cells in MBEN differs between internodular and nodular tumor compartments. a** Representative scan of MB299 shows gene expression of the neuronal differentiation marker *NRXN3* (purple) and *LAMA2*, a marker gene that is strongly expressed in stromal and astrocytic(-like) cells (green). The spatial expression of these two genes is sufficient to reconstruct the bicompartmental structure of MBEN. **b** UMAP projection showing the clustering of single cells derived from molecular cartography, which reconstructs all stages of MBEN development that were identified with snRNA-seq alongside none-malignant cell types (*n* = 4 patients/92,666 cells). **c** Heatmap depicting marker genes of the different clusters from **c**. **d** Single cells were mapped onto the full MBEN snRNA-seq dataset, with clear evidence that non-malignant cell types could be assigned to astrocytes, fibroblasts, and microglia, respectively. **e** UMAP-plots showing the expression of the astrocytic marker genes *NPAS3*, *CD44* and *LAMA2*, which are expressed in astrocytic and a fraction of early CGNP-like cells ((*n* = 4

patients/ 92,666 cells each). The feature plot shows the respective expression in MBEN clusters from snRNA-seq data. **f** Representative scan of MB266 in which single cells were projected back into the spatial space. By integrating all cell types derived from single cell construction, the bicompartmental structure of MBEN is reconstructed. The color code is equivalent to **b**. Mapping of **g** proliferating, early CGNP-like, **h** intermediate, migrating, and **i** neuronally differentiated MBEN-cells. Whereas early CGNP-like cells are restricted to the internodular compartment, neuronally differentiated cells form the tumor nodules. Migrating MBEN-cells are spread throughout both compartments. **j** Expression patterns of *NRXN3* (marker for differentiated neuronal-like MBEN cells) and *AQP4* (astrocyte marker) in a representative scan of MB266. **k** Astrocytic(-like) cells (red) are located at the transition border of the nodular compartment, which is formed by differentiated, neuronal-like cells (green). Scale bars in **f**–**k** 400 μm. Source data are provided as a Source Data file.

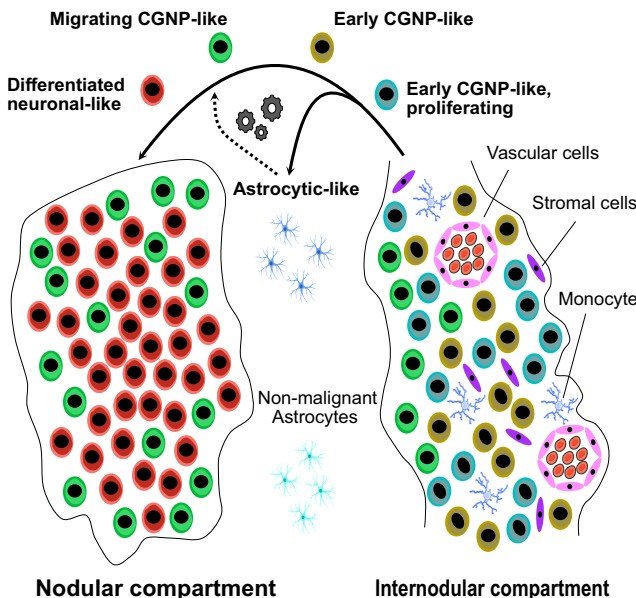

**Fig. 8 | MBEN histology recapitulates physiological CGNP development.** Graphical summary that visualizes the spatial and transcriptomic structure of MBEN and its compartments on the single cell level. The images "erythrocyte" (by Servier, licensed under https://creativecommons.org/licenses/by/3.0/) from the database https://bioicons.com/ as well as "Blue Astrocyte" (by Andrew Hardaway, licensed under https://creativecommons.org/licenses/by/4.0/) and "Microglia Resting" (by John Chilton, licensed under https://creativecommons.org/licenses/by/4.0/) from the database https://scidraw.io/ were used.

quantification and quality control using the Qubit fluorometric quantification (ThermoFisher Scientific) and TapeStation system (Agilent) was performed, and the cDNA libraries were sequenced on a Nextseq550.

### Analysis of snRNA-seq raw data
Initial processing of 10X data (reads alignment, counts computation per cell) was performed with CellRanger v3 pipeline. The SMARTSeq V 2.3 (SS2.5) data was processed per sample representing a cell with bulk RNA-seq derived procedure as described previously[86]. Briefly, single nucleus Smart-seq2 reads alignment was performed with STAR v 2.4.1d, gene expression counts were computed for each cell per sample with Subread 1.6.4 tool, and afterwards merged into a gene expression matrix via custom Python scripts within the python 2.7 environment. For both protocols hg19 human genome reference combined with gencode v19 gene annotation were used. For gene expression counts computations exons were merged with introns due to single nuclei protocol specificity[92].

Computed cell gene expression counts were further analyzed per sample with Seurat v4.0.3 toolkit[93]. Filtering control limits (minimum number of molecules/genes) were identified based on the quality control inspection. The doublets detection was performed with DecontX v1.1.0 tool[94]. Copy number profiling was performed with InferCNV v1.10.1[36,95]. Signatures for Ca2+ and BMP signaling were constructed using the module score function within Seurat. Gene ontology analysis was performed using the PANTHER classification system (Annotation version: 2021-02-01) with DEGs per cluster as input (http://www.pantherdb.org/)[63]. The results were visualized with the REVIGO web application by clustering based on semantic similarities[96].

Initially 10X and SS2.5 datasets were analyzed separately; their identified clusters were merged into pseudobulk blocks and compared via correlation. Non-tumor cells were further annotated based on inspection of clusters from merged datasets: the clusters containing cells from multiple samples were considered as normal while consisting from one sample on 95% as a tumor. Afterwards, the datasets (full and tumor only) were merged using Harmony v0.1.0 package[31] considering both sample and protocol as batch effects. This approach was validated using an entropy analysis that showed ROGUE-scores[97] around 0.9 for all clusters, indicating high internal cluster stability (ROGUE v1.0 package). Trajectories were identified via Slingshot v2.0.0 on the full merged cohort and Monocle2 v2.20.0 on target samples as described in the respective studies[50,98]. The similarity of tumor cells to normal cerebellum cell types was analyzed using the SingleR v1.8.1 package[32].

In order to test for associations between different cell states and the variables age of onset, sex, and relapse status, the percentage of each cell state per patient was calculated. Subsequently, Spearman correlation was used to test for any significant associations.

For Transcription factor (TF) activity estimation the python version of DecoupleR v1.3.0 package was used[99]. The needed conversion from Seurat objects to AnnData objects was done with the SeuratDisk library (https://mojaveazure.github.io/seurat-disk/). TF activity estimation was performed based on DoRothEA which is a comprehensive prior knowledge resource containing curated TFs and their targets[66]. This network was derived from the OmniPath database[100] via DecoupleR (R package: OmnipathR v3.7.0). In DoRothEA, each TF-target interaction includes a confidence level annotation ranging from A to E based on the supporting evidence where A is the highest confidence level and E is the lowest. For this analysis, TF-target pairs coming from the three highest confidence levels (i.e., A, B and C) were used to create a predictive model for TF activity estimation. Here, we apply a multivariate linear model to every cell in our samples to estimate the log-transformed gene expressions using weights assigned to the interactions between TFs and genes. After the model is trained, the resulting t-values of the slopes serve as scores. A positive score indicates an active pathway, while a negative score indicates an inactive pathway. The resulting activities were

summarized per cluster by their mean with the summarize_acts function. The minimum standard deviation was set to zero to retrieve all results.

In order to analyze cell-cell communication between cell types, we conducted a ligand-receptor analysis with the Ligand-receptor Analysis framework v0.1.6 (LIANA). LIANA combines different prior knowledge resources and different in silico methods by taking the consensus of the ligand-receptor predictions. The tool was used with default settings as described in the tutorial at https://saezlab.github.io/liana/articles/liana_tutorial.html[58]. This results in the usage of the methods CellPhoneDB, Connectome, log2FC, NATMI, SingleCellSignalR and CellChat. A consensus between these methods was reached by applying the robust rank aggregate method. Five expertly curated CCC resources derived from the Omnipath database, namely CellPhoneDB, CellChat, ICELLNET, connectomeDB2020 and CellTalkDB, were used as prior knowledge.

## Microdissection

Microdissection with subsequent RNA isolation was performed on FFPE-derived histological slides from 26 MBEN patients as previously described[101]. The main analysis of microdissected samples was performed as for standard bulk RNA-sequencing[86]. The clustering was performed from normalized gene expression counts with a focus on top 500 highly variable genes. Differentially expressed genes between nodular and inter-nodular blocks were detected via the limma package, with considering the tumor sample for batch effect correction and using minimum limits of 0.5 for log2 fold change, and 0.05 for adjusted *p*-values. Further, nodular and inter-nodular detected groups of differentially expressed genes were used as a reference for Gene Set Variation Analysis[71] on the genes identified as markers of clusters in MBEN single cell data. The MuSiC deconvolution method (R package: MuSiC v0.9) was applied on the bulk internodular and nodular RNA-seq datasets using the snRNA-seq derived MBEN cell stages as the ref. 72. Afterwards, the proportions of predicted cell types between groups were compared using two sided *t*-tests.

## Immunohistochemistry

IHC was conducted on 4-μm thick FFPE tissue sections mounted on adhesive slides followed by drying at 80 °C for 15 min. For IHC analysis, a mouse monoclonal GFAP (GA5; Cell signalling, catalogue number #3670) and Nestin (MAB5326; Merck/Sigma-Aldrich) antibodies were applied. IHC was performed with an automated immunostainer (Benchmark; Ventana XT) using antigen-retrieval protocol CC1 and a working antibody dilution of 1:2000 for GFAP and 1:200 Nestin with incubation at 37 °C for 32 min. The antibody against Nestin has been validated by the manufacturer (Merck/Sigma-Aldrich) by detecting Nestin in a Western Blot Analysis on 10 μg of Huvec Lysates (Dilution: 1:1000) as well as Immunohistochemistry on formalin fixed, paraffin embedded (FFPE) tissue (1:200) and Immunocytochemistry on formaldehyde fixed cultured cells (1:200) as described and stated in the manufacturer's description (https://www.merckmillipore.com/DE/de/product/Anti-Nestin-Antibody-clone-10C2,MM_NF-MAB5326). The antibody against GFAP has been validated by the manufacturer (Cell Signaling) by detecting GFAP in a Western Blotting Analysis (Dilution: 1:1000), Immunohistochemistry on FFPE- (1:50 - 1:200), and FF-tissue (1:400 - 1:800), Immunocytochemistry (1:400 - 1:800), and Flow Cytometry (1:400 - 1:1600) as described and stated in the manufacturer's description (https://www.cellsignal.com/products/primary-antibodies/gfap-ga5-mouse-mab/3670).

## smRNA-FISH (RNAScope)

**Histological sample preparation.** Sectioning of fresh/frozen tissue derived from patients MB266, MB295 and MB299 was performed at

−20 °C on a cryostat (Leica) and 8-10 μm sections were mounted on Superfrost Plus slides (ThermoFisher). Cryosections were stored at −80 °C until further use.

**Virtual H&E staining and probe hybridization.** 12-plex single molecule RNA-FISH (smRNA-FISH) was performed using the RNAScope HiPlex assay (ACDbio/biotechne) as described in the 'RNAScope HiPlex Assay User Manual (324100-UM)' with minor adaptions (Supplementary Data 7). Briefly, sections were fixed in 4% paraformaldehyde (PFA) for 60 min, washed two times with PBS and dehydrated in Ethanol. For virtual H&E staining (PMID: 29531846), sections were stained with Eosin (Sigma, 1:10 diluted in 0.45 M Tris acetic acid, pH=6) for 1 min at room temperature, washed in H₂O and incubated for 15 min in 4x SSC buffer. Sections were then stained with DAPI for 30 sec and mounted in Prolong Gold Antifade (ThermoFisher). After the first virtual H&E imaging round (R0), the coverslip was removed by incubation in 4x SSC buffer for 15-30 min. Afterwards, sections were washed in PBS once and again dehydrated in Ethanol. Sections were then treated with Protease IV (ACDbio) for 30 min at RT, washed 2x with PBS and incubated with transcript-specific (Supplementary Data 7) and amplifier probes according to the manufacturer´s instructions. Between imaging rounds, fluorophores of the previous imaging rounds were cleaved to enable consecutive rounds of imaging, with each round targeting a new set of transcripts. Up to four transcripts were labeled per imaging round by Alexa488, Atto550, Atto647 and Alexa750 fluorescent dyes. For MB266 and MB299, four transcripts (+ DAPI) were imaged in three imaging rounds (R1-R3). For MB295, three transcripts (Alexa488, Atto550, Atto647 and DAPI) were imaged in four imaging rounds using the 'RNAScope HiPlex Alternate Display Module' (R1-R4).

**Microscopy.** smRNA-FISH images were acquired on an Andor Dragonfly confocal spinning-disk microscope equipped with a CFI P-Fluor 40X/1.30 Oil objective. The region of interest was selected based on the DAPI signal and 50 z-slices were acquired with a step size of 0.4 μm (20 μm z-range) per field of view (FOV). Lasers and filters were set to match fluorescent properties of DAPI and above mentioned dyes. Tiles were imaged with a 10% overlap to ensure accurate stitching.

**Image analysis.** Pre-processing of images was performed in ImageJ. Images were projected in z using Maximum Intensity Projection. Illumination correction was performed on the DAPI and Eosin images for virtual H&E visualization using the BaSiC plugin[102]. Image tiles were stitched using 'Grid/Collection stitching' and registered afterwards based on the DAPI signal using 'Register Virtual Stack Slices' using the Affine feature extraction model and the Elastic bUnwarpJ splines registration model. Virtual H&E transformation of DAPI and Eosin staining was performed in R using the EBImage tool and custom script[103]. Nuclei were segmented by the Cellpose v 2.0.5 python tool using the 'cyto' model and a model match threshold of 1.5[104]. Nuclei outlines were exported to ImageJ and transformed to ROIs using the ROImap function of the LOCI plugin. Spot detection was performed using the RS-FISH plugin in ImageJ (10.1101/2021.03.09.434205) with Sigma set to 0.93. The threshold for spot detection was adapted for each patient and fluorophore individually (range: 0.004 – 0.0086). Transcripts were assigned to nuclei using a custom KNIME script that overlaps the DAPI and spot signal and counts spots per nuclei. Nuclei metadata including x/y coordinates and other features were extracted using the 'Segment Features' node in KNIME.

**Downstream analysis.** Raw transcript count matrices were generated from KNIME outputs in R using custom script. Transcript counts for *LRRTM4* and *FOS* were removed from the analysis due to strong

discrepancies to snRNA-seq data. smRNA-FISH data was further analyzed with the R package Seurat v4.0.3[105]: Cells were filtered by transcript count (< 5 and >100 transcripts) as well as by nuclei size (< 90 and > 2000 pixels). smRNA-FISH data was normalized using scTransform v0.3.2[106], visualized using principal component analysis and clustered using the Louvain algorithm. Clusters with similar marker gene expression and high correlation of their average gene expression profiles were merged and assigned to cell types according to their marker expression as detected in snRNA-seq data. Further spatial analysis and visualization of smRNA-FISH data including spatial networks was performed using the Giotto v1.1.1 tool in R[107]. Cell type-specific nuclei mask images were generated using KNIME and visualized with napari (https://napari.org).

### Molecular Cartography (Resolve Biosciences)

To ensure that both spatial transcriptomic methods were comparable, smRNA-FISH marker genes were included in our MB panel, alongside marker genes of the developing cerebellum, oncogenes, and markers of non-malignant cell types. OCT-embedded samples were cryosectioned as described above into 10 μm sections onto an MC slide. Fixation, permeabilization, hybridization and automated fluorescence microscopy imaging were performed according to the manufacturer's protocols (Molecular preparation of human brain (beta), Molecular coloring, workflow setup) except for a few adaptations indicated below. Briefly, slides were thawed at room temperature and dried at 37 °C. Subsequently, the MC observation chamber was assembled by attaching sticky wells (8-well) to the MC slide. Sections were fixed, permeabilized, rehydrated and treated with TrueBlack (Biotium) autofluorescence quencher. In contrast to the MC protocol provided by Resolve Biosciences, we diluted the quencher 1:20 as specified by Biotium. Next, the sections were thoroughly washed and primed before the specific probes against our genes of interest were hybridized at 37 °C overnight. The probe sequences were designed by Resolve Biosciences' proprietary algorithm and are hence not listed here. After hybridization, the sections were washed, and the MC observation chamber was transferred to the MC machine for eight automated iterations of coloring and imaging to decipher the transcript localization of the 100 different genes of interest in the tissue[108]. Therefore, regions of interest (ROIs) were selected for each section (MB263, MB266, MB295, MB299) based on a brightfield overview scan. In the last imaging round, Nuclei were stained with DAPI yielding a reference image for nuclei segmentation. After the run, the MC software performs registration of the raw images, assigns transcripts to the combinatorial color codes detected and combines individual tiles to ROI panoramas. The outputs are text files containing the transcript coordinates in 3D as well as maximum projections of DAPI images for each ROI.

After the MC run, we stained the tissues with an anti-NCAM antibody and eosin for virtual H&E images as described above. All following steps are carried out at room temperature and washing was always done 3x with PBS unless specified otherwise. Sections were washed, fixed in 4% paraformaldehyde (PFA) for 30 min, washed again, permeabilized with 0.2% Triton X-100/PBS for 12 min and washed again. Next, the sections were blocked for 1 h in 10% goat serum/PBS and then incubated with a 1:200 dilution of the primary antibody (mouse anti-NCAM/CD56 from ThermoFisher Scientific, MA1-06801, lot: 17425) in 10% goat serum/PBS for 1 h. After washing 3×5 min with 0.002% NP-40/PBS, the sections were incubated with a 1:300 dilution of the secondary antibody (goat anti-mouse Alexa568) in 10% goat serum and washed again before and after a 15 min incubation with 5 μM DAPI. After washing 1x with double-distilled water, the sections were incubated with Eosin mix (1 vol Eosin Y Sigma HT110216 to 9 vol Tris Acetic acid buffer 0.45 M, ph6) for 1 min. After ten washes with double-distilled water, MC

imaging buffer was added to the sections and they were imaged on the Andor Dragonfly microscope described above with the following setup: 60x objective, 41 z-slices at 0.3 μm distance, 10% overlap between tiles for stitching, EM gain 200, excitation at 405 nm (DAPI), 488 nm (Eosin) and 647 nm (immunofluorescence) with appropriate emission filters.

DAPI images were flatfield-corrected using a separately recorded flatfield and darkfield image. Stitching and registration of the Dragonfly DAPI image to the MC DAPI image (from round 8) and the calculation of the virtual H&E images was done as described above.

The detection of cell boundaries was performed with QuPath v0.3.2[109]. Afterwards, gene expression counts were computed per cell and extracted using Resolve Bioscience plugin in the ImageJ2 toolkit.

Formed gene expression matrix analysis (filtering, clustering, visualization) was performed with Seurat toolkit[93]. The samples were merged using the Harmony v0.1.0 R package with batch effect adjustment[31]. Cell states and types were annotated by comparison to the snRNA-seq dataset and by visual inspection of marker genes. Clusters which only differed marginally in gene expression were combined. Additional spatial analysis (closest cell connections detection) was performed with Giotto v1.1.1 toolkit analogous to the analysis of the smRNA-FISH dataset[107].

### Statistical analysis and visualisations

The Kaplan-Meier-method was applied to analyze and visualize progression free and overall survival. The respective analysis were performed using the R packages survival v3.2.11 (https://github.com/therneau/survival) and survminer v0.4.9 (https://github.com/kassambara/survminer). Descriptive statistics and visualizations were conducted using the R base and ggplot2 v3.3.3 packages.

### Writing

The Large Language Model (LLM) ChatGPT by OpenAI (July20 version) was used to assist in the formulation of the abstract of this paper.

### Reporting summary

Further information on research design is available in the Nature Portfolio Reporting Summary linked to this article.

## Data availability

The snRNA-seq and bulk-sequencing (RNA, microdissected) data have been deposited in GEO database and is available under the combined accession number GSE239854. All raw images and processed data after cell segmentation from spatial transcriptomics experiments have been deposited at the BioImage Archive and can be accessed under the accession numbers S-BIAD825, S-BIAD826. In addition, the raw counts for Molecular Cartography have been uploaded to GEO (accession number: GSE247736). The DNA methylation data copy number profiles and DNA sequencing mutation results were integrated from the corresponding medulloblastoma molecular landscape study deposited at European Genome-Phenome Archive under accession number EGAS00001001953[12]. All raw data acquired by DNA methylation profiling is available via GEO (accession number: GSE247741). The fastq files from DNA sequencing are available via SRA (SRA-ID: 473652, BioProject: PRJNA1044021). The remaining data are available within the Article, Supplementary Information. Source data are provided with this paper.

## Code availability

Scripts for processing the raw data and generating the figures are available via the GitHub repository: https://github.com/kokonech/MBEN_snData_analysis and have been uploaded to Zenodo (DOI:10.5281/zenodo.10047009)[110].

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

## Acknowledgements

We thank the High-Throughput Sequencing Unit of the Genomics and Proteomics Core Facility, German Cancer Research Center (DKFZ) for providing excellent services regarding all sequencing experiments. This project received funding by a grant of the European Research Council (ERC) to S.M.P. under the European Union's Horizon 2020 research and innovation program as part of the initiative BRAIN-MATCH (Grant agreement ID: 819894). D.R.G. was supported with personal grants by the German Academic Scholarship Foundation (Studienstiftung des Deutschen Volkes) and the Mildred Scheel Doctoral Fellowship program of the German Cancer Aid (Deutsche Krebshilfe). A.S.R. and J.S.-R. were supported by the German Federal Ministry of Education and Research (BMBF) through the grant CompLS DeepSC2 (031L0269B). The authors gratefully acknowledge the data storage service SDS@hd supported by the Ministry of Science, Research and the Arts Baden-Württemberg (MWK) and the German Research Foundation (DFG) through grant INST 35/1314-1 FUGG and INST 35/1503-1 FUGG. D.R.G. and K.W.P. are thankful to the non-profit foundation Ein Kiwi gegen Krebs for their support. The development of spatial transcriptomics protocols was supported by the Baden- Württemberg Stiftung (project MET-ID41-STARFISH) and by the HMLS Explorer program of the University of Heidelberg (project START-HD). H.S. gratefully acknowledges funding from the Ministry of Science, Research and the Arts Baden-Württemberg (MWK) during the Science Data Centre project BioDATEN.

## Author contributions

Acquisition, analysis, and interpretation of data: D.R.G., K.O., A.R., S.T., K.K.M., H.S., M.P.G., P. J., J.S., B.S., A.S.R., K.B., S.S., M.B., F.G., K.E., J.S.-R., D.T.W.J., S.C.C., E.F., D.K., J.-P.M., K.R., A.K., S.M.P., K.W.P. Design and conceptualization: D.R.G., K.O., A.K., S.M.P., K.W.P. Writing of the original draft: D.R.G., K.O., A.K., S.M.P., K.W.P. Substantial revisions and feedback to the draft: All authors.

## Funding

## Competing interests

S.T. is an employee of the company Resolve BioSciences GmbH. JSR reports funding from GSK, Pfizer and Sanofi and fees/honoraria from Travere Therapeutics, Stadapharm, Astex, Pfizer, Owkin and Grunenthal. No other author declares a conflict of interest.

## Additional information

[1]Hopp-Children's Cancer Center Heidelberg (KiTZ), Heidelberg, Germany. [2]Division of Pediatric Neuro-oncology, German Cancer Consortium (DKTK), German Cancer Research Center (DKFZ), Heidelberg, Germany. [3]Department of Pediatric Oncology, Hematology, and Immunology, Heidelberg University Hospital, Heidelberg, Germany. [4]Division of Chromatin Networks, German Cancer Research Center (DKFZ) and Bioquant, Heidelberg, Germany. [5]Heidelberg University, Faculty of Medicine, and Heidelberg University Hospital, Institute for Computational Biomedicine, Bioquant, Heidelberg, Germany. [6]Department of Biological Engineering, Massachusetts Institute of Technology (MIT), Cambridge, MA, USA. [7]Faculty of Biosciences, Heidelberg University, Heidelberg, Germany. [8]Department of Electrical and Electronics Engineering, İskenderun Technical University, Hatay, Turkey. [9]Single-cell Open Lab, German Cancer Research Center (DKFZ), Heidelberg, Germany. [10]Department of Radiological, Oncological and Anatomo-Pathological Sciences, Sapienza University of Rome, Rome, Italy. [11]Istituto di Ricovero e Cura a Carattere Scientifico (IRCCS) Neuromed, Pozzilli, Italy. [12]Division of Pediatric Glioma Research, German Cancer Research Center (DKFZ), Heidelberg, Germany. [13]Wolfson Childhood Cancer Research Centre, Translational and Clinical Research Institute, Newcastle University Centre for Cancer, Newcastle upon Tyne, UK. [14]Department of Biochemistry and Cellular Biology, National Center of Neurology and Psychiatry (NCNP), Tokyo, Japan. [15]Edythe Broad Institute of MIT and Harvard, Cambridge, MA, USA. [16]Department of Neuropathology, Institute of Pathology, Heidelberg University Hospital, Heidelberg, Germany. [17]Clinical Cooperation Unit Neuropathology, German Cancer Research Center (DKFZ), German Consortium for Translational Cancer Research (DKTK), Heidelberg, Germany. [18]Present address: Resolve BioSciences GmbH, Monheim am Rhein, Germany. [19]These authors contributed equally: David R. Ghasemi, Konstantin Okonechnikov. [20]These authors jointly supervised this work: Andrey Korshunov, Stefan M. Pfister, Kristian W. Pajtler. ✉e-mail: andrey.korshunov@med.uni-heidelberg.de; s.pfister@kitz-heidelberg.de; k.pajtler@kitz-heidelberg.de

