## [Peer Review File · Nature Communications]

Compartments in medulloblastoma with extensive nodularity are connected through differentiation along the granular precursor lineage.Reviewers' Comments:

Reviewer #1:

Remarks to the Author:

Ghasemi et al. investigated medulloblastoma with extensive nodularity (MBEN) using single cell profiling. They found that the internodular MBEN compartment is composed of proliferating early cerebellar granular neuronal precursor (CGNP)-like tumor cells, stromal, vascular, and immune cells. The nodular compartment contains postmitotic, neuronally differentiated MBEN cells. The authors also identified astrocyte-like tumor cells that branch off the main CGNP developmental trajectory, and are located near migrating late CGNP-like and postmitotic cells with neural differentiation. The study connects the spatial tissue organization in MBENs to their developmental trajectory from proliferation to differentiation.

Some comments that I hope can help the authors in further improving their study:

1. The findings of tumor-derived astrocyte-like cells are intriguing. Is there evidence of high activation of the SHH pathway in these cells? Can the authors analyze mutations in SHH-pathway genes using smart-seq data sets that sequence all exons? Do the tumor astrocytes have distinct signatures differentiating them from normal astrocytes? If so, can these be verified using publicly available bulk RNA-seq data from MBENs? Is it possible that these tumor astrocytes are radial glia or another type of basal progenitor?
2. The authors mention that three children in the cohort had relapse. Is there any correlation with specific cell types/signatures and relapse?
3. Is there any association between age/sex and differentiation of neurons (e.g. more differentiation in younger patients)?
4. The authors mention a mesenchymal signature, but it is not described in detail in the manuscript.
5. For the DoRothEA analysis, how was the measurement of changes in TF activity performed?
6. Can the authors provide more information on microglia/macrophages observed in the snRNA-seq datasets? Are there molecular signatures associated with pro- and anti-inflammatory markers, resident microglia/bone marrow derived macrophage etc?

Reviewer #2:

Remarks to the Author:

The current study presents an innovative integration of single cell and spatial transcriptomic analysis of medulloblastoma with extensive nodularity, offering a valuable opportunity to examine spatial dependencies. The manuscript is well-written, though some adaptations to the figures would enhance interpretation (as noted below). The analysis and interpretations are cutting-edge. Prior to publication, several aspects should be considered:

1. The CNV alterations in astrocytes were only detected in one patient, and the annotations of patients are missing in the CNV heatmap for the "non-malignant" cells.
2. The approach of using non-integrated data to identify tumor cells may be challenging, as batch horizontal integration should remove the confounding factor of technical differences such as the use of two sequencing platforms. Given the significant difference in sequencing depth between the 10X and SMART-seq protocols, it would be insightful to examine the distribution of these platforms across the identified malignant and non-malignant cells.
3. The approach should be quantified through an entropy examination of individual clusters.

4. The term "tumor astrocytes" may not be appropriate, as the scientific literature usually refers to these cells as "tumor-associated astrocytes" or "reactive astrocytes." If malignant cells mirror differentiation trajectories of healthy cells, they are typically described as "astrocytic-like" (Neftel et al., 2019), as the authors have used the terms "CGNP-like" or "neuronal-like."
5. The hierarchical states analysis using the "sling-shot" approach is not straightforward, and additional pseudotime tools such as STREAM or Monocle2 would provide a clearer demonstration of hierarchical ordered trajectories.
6. The cell-cell interaction analysis would benefit from the integration of spatial data, such as the cellular components of each spatial niche. For example, the authors later describe a juxtaposition of neuronal-like and migrating CGNP-like cells, and it would be more appropriate to analyze these interactions in accordance with relative spatial distance.
7. The exclusion of non-malignant cells from the analysis is not clear.
8. The integration of single cell data with the microdissection RNA-seq data would improve the cell type distribution analysis, for example using MuSiC or other tools.
9. The figure 6 presentation should use consistent colors for cell types and maintain a consistent order of cells in the legend between plots.
10. For a better understanding of cellular co-localization, it would be useful to present integrated data of all three patients in one graph, rather than three individual graphs.
11. The microenvironment with respect to cellular distribution between spatial and single cell appears to differ, and this difference should be quantified and highlighted.
12. An extended analysis of cellular co-localization and potential receptor-ligand interactions would offer novel insights on the importance of spatial niches, similarly to the RNA-scope data.
13. The quality data regarding the UMI per cell and the detected genes per cell for both single cell and spatial data are missing. Information about the 100-plex Molecular Cartography platform, including whether it was customized, is also not available.

Reviewer #3:

Remarks to the Author:

Medulloblastoma with extensive nodularity (MBEN) is a unique histological subtype of MB characterized by its appearance with distinct compartments. It mainly arises in infants and in very young children, and it is associated with a good prognosis. However, some patients relapse or present disease progression.

In the current study "Compartments in medulloblastoma with extensive nodularity are connected through differentiation along the granular precursor lineage" from Ghasemi et al., the authors investigate the biology underlying the histological compartments of MBEN and how they are interconnected, as it could provide insight into the varied aggressiveness of the disease and the degree of tumor differentiation.

The authors have performed a very thorough analysis using a multimodal transcriptomic approach that improves our understanding of MBEN and would be a fantastic research article for Nature Communications. Their research reveals the intratumoral heterogeneity present within the nodular and internodular compartments, identifying a spectrum of cell stages with a developmental trajectory that mimics the differentiation of CGNP-lineage during cerebellar development, as well as the presence of MBEN-associated tumor astrocytes.

Prior to consideration for publication, the following comments should be considered/addressed.

Comments:

- Two methods of single nucleus RNA-sequencing were used to analyze the MBEN patient samples: 10X V2 3'- (n = 9) and the SMARTseq V2.5-protocols (n = 6). These data are later integrated in the analysis.

o Is there a reason why 3/9 samples are missing or were excluded from the analysis? How did this affect the number of cell populations identified per patient sample? For example, the number of tumor astrocytes (SFig3G)?

- How many CNVs (CNV-signature, line 187) specific to the tumor astrocytes were identified? In how many patients are they identified? In line 195 it is indicated that tumor astrocytes were identified in all samples, however, this is not very clear from the graphs. In Fig 2C it seems that there is one big region with CNV gain in chr2 specific to the tumor astrocytes, but this one is only observed in 1/9 patients (blue) from the panel underneath. Also, in S.Fig 3G there is a pie chart indicating the number of astrocytes per sample suggesting they were identified in all cases although in varied numbers (min 3, max 208). However, in Figs D-E, F-G it seems that they are only present for some patients.

o A table with the list of CNVs (CNV-signature) specific to the tumor astrocytes should be provided, particularly indicating which ones were identified for each patient.

- Line 254. How was the pseudo-temporal ordering performed?

- Fig 4. Could some of the receptor-ligand-interactions identified be validated?

- Fig 5 A. The cut off (logFC, adjusted p-value) used to identify the subset of differentially expressed genes among nodular and internodular compartments should be indicated.

- Fig 5 B and C. Statistical analysis comparing the expression levels of TMEM108 and TRIM9 should be included.

- Spatial transcriptomics correlates cell stages in MBEN to its histologic compartments. "For RNAscope, twelve genes were chosen as markers for the different snRNA-seq clusters, and also considering biological significance". How were these genes chosen? S. Table 7 indicates the gene probes used for the 12 genes.

o Additional information should be included indicating: from which population were each chosen, the reason why they were included (explanation and biological significance +/- highest expression and unique to the cluster?) and the 2 genes that were excluded.

- The authors hypothesize about the clinical relevance of their findings. Additional analysis could be considered, including:

- o How could this finding be translated into the clinical setting? From the cohort of 26 samples analyzed (microdissected tissue), IHC with the 2 set of genes representative of each compartment could be performed (TMEM108/TRIM9; NRX3/LAMA2).

- o Does it have prognostic implications? The proportion of differentiated vs non-differentiated populations could be calculated for the cohort of 26 samples analyzed and integrate the results with survival analysis. What are the differences between patients with a better outcome compared to those that progress or are metastatic?

- Data availability: Data should be made publicly available, and a data availability statement should be included.

Minor comments:

Fig. 1 Method spectrum and molecular features of the MBEN patient cohort.

- Fig 1B. Labelling of axis is missing for the t-SNE plot

- Fig 1C. It is not very clear how many patients present the alteration. The X axis indicates patients 1-5 for the cohort of 9, and there are 2 squares to indicate 1 patient. Is the x axis numbering wrong?

- Fig 1D. Scale magnification should be included. It would also be helpful that an example of each compartment is highlighted for the non-experts in histopathology.

Fig. 2 SnRNA-seq reveals a subset of MBEN-cells with similarity to astrocytes.

- Fig. 2C. It is a bit confusing to understand whether the colors represent patient samples or cell populations. The legend corresponding to the top panel should be moved up so it is clear that the first panel focusses on cell populations while the second one on patient samples.

- Fig. 2E. Letter E is duplicated in the figure legend, one is C.

Fig4. Biological processes that regulate MBEN differentiation mimic CGNP development.

- specify which population each cluster represents

Fig 6. Spatial transcriptomics map developmentally distinct cell stages to spatially distinct tumor compartments.

- Add scale in tissue sections

Fig 8. MBEN histology recapitulates physiological CGNP development.

- Missing figure legend

References

- Format issue in reference 56, it includes 5 weblinks.

Reviewer #1 (Remarks to the Author): Expert in snRNA-seq, neuro-oncology, and omics for brain tumours

Ghasemi et al. investigated medulloblastoma with extensive nodularity (MBEN) using single cell profiling. They found that the internodular MBEN compartment is composed of proliferating early cerebellar granular neuronal precursor (CGNP)-like tumor cells, stromal, vascular, and immune cells. The nodular compartment contains postmitotic, neuronally differentiated MBEN cells. The authors also identified astrocyte-like tumor cells that branch off the main CGNP developmental trajectory, and are located near migrating late CGNP-like and postmitotic cells with neural differentiation. The study connects the spatial tissue organization in MBENs to their developmental trajectory from proliferation to differentiation.

Some comments that I hope can help the authors in further improving their study:

1. The findings of tumor-derived astrocyte-like cells are intriguing. Is there evidence of high activation of the SHH pathway in these cells?

We thank reviewer #1 for these interesting questions and helpful suggestions. There is in fact evidence of SHH pathway-activation in astrocytic-like malignant cells. As shown in **Fig. 3c** and **Suppl. Fig. 4a-c**, several key SHH-pathway genes (e.g., *HHIP*, *BOC*, *GLI2*) are upregulated in this cell population. We have added these observations in the manuscript.

p. 9, l. 2: “In addition, astrocytic-like cells showed upregulation of stromal and of early cell stage markers (*LAMA2*, *SOX2*, *SOX9*) as well as SHH-pathway members (e.g., *HHIP*, *BOC*, *GLI2*) (**Fig. 3c**, **Suppl. Fig. 4a-c**), but no significant proliferative activity.”

Can the authors analyze mutations in SHH-pathway genes using smart-seq data sets that sequence all exons? Do the tumor astrocytes have distinct signatures differentiating them from normal astrocytes? If so, can these be verified using publicly available bulk RNA-seq data from MBENs?

The reviewer is raising a very important point. Due to the comparably lower number of sequenced cells, there were only few cells with an astrocytic phenotype present within the SMARTseq V2.5 dataset. As suggested, we analyzed these cells with regard to somatic mutations, which were derived from bulk sequencing data. We were able to identify mutations in the *SMO* gene, a known SHH-pathway member, in three astrocytic cells from one case (MB266) (**Fig. 2d – f**). The clear copy number variations in astrocytic cells, which we already described in the first version of our manuscript, occurred in another sample (MB295) (**Fig. 2c**). Thus, these two independent findings further confirm our initial observation that astrocytic cells in MBEN are at least partially of malignant nature and have therefore been included both in **Fig. 2** as well as the text of the respective paragraph.

As described in the paper, we observed that astrocytic cells clustered either in a mixed cluster of non-malignant cells or within patient-specific malignant populations when analyzed without correcting for patient-specific batch effects, and in the absence of any form of *in vitro* or *in vivo* MBEN models, we used these findings to estimate which cells were non-malignant and malignant, respectively (**Fig. 2g**, **Suppl. Fig. 4h – j**). In addition, as shown in **Fig. 3e**, astrocytic-like tumor cells showed transcriptional similarity to murine astrocytic-like tumor cells from a study by Yao *et al.* (Cell, 2020). In order to distinguish non-malignant and malignant astrocytic cells within the MBEN tumor microenvironment, high-throughput single cell DNA-sequencing methods will be necessary that allow to safely identify SNVs from a very large number of cells. However, these methods are just beginning to emerge and

are not widely available as of yet. In order to generate more insights on the nature of astrocytic cells in MBEN, we stained all samples from our FFPE-cohort for which material was still available (n = 12) with GFAP and Nestin as a marker for a reactive astrocytic cell state. In line with our results from the spatial transcriptomics experiments, both markers generally colocalized in the periphery of the nodular compartments (**Suppl. Fig. 16a - d**). Interestingly, we observed marked differences between the more differentiated MBEN subtype TCL2, which made up the majority of samples in our snRNA-seq./spatial transcriptomics cohort, and the more aggressive, dedifferentiated subtype TCL1 (Korshunov *et al.*, 2020), in which both markers were less prominently detected (**Suppl. Fig. 16a – d, Suppl. Tbl. 11**). To follow this up, we analyzed the expression of *GFAP* and *Nestin* in MBENs as compared to DNMBs. Whereas these markers were significantly correlated in MBEN, this was not the case in DNMBs, suggesting that there are differences between the astrocytic cell population in these related, but different tumors (**Suppl. Fig. 16e,f**).

These new results have been included in the manuscript:

p. 6, l. 17: “Since SMARTseq V2.5 generates whole transcriptome reads that can be used for the analysis of single nucleotide variants (SNVs), we investigated whether we could identify cells with both an astrocytic phenotype and pathogenic SNVs. Strikingly, a small number of astrocytic cells (n = 3) from sample MB266 indeed harbored the same mutation within the SHH pathway gene *SMO* as detected using bulk sequencing, further confirming our initial findings that a fraction of astrocytic cells in MBEN is derived from malignant precursors (**Fig. 2d - f**).”

p. 19, l. 35: “In order to further validate the localization of astrocytic cells in MBEN, we stained a cohort of MBENs with available FFPE-derived tissue (n = 12) astrocytic marker GFAP. In full concordance with our spatial transcriptomics data, GFAP was expressed in the peripheral zone of the nodular MBEN compartment (**Suppl. Fig. 16a – d**). GFAP-positive cells showed various appearances, ranging from mature astrocytes with numerous processes to small, primitive-like cells. Interestingly, reactive astrocytes as characterized by Nestin-expression, were only found in 1/5 cases of the TCL1-, but 6/7 cases of the TCL2-subtype (**Suppl. Fig. 16a – d, Suppl. Tbl. 11**). To follow this up, we analyzed the expression of GFAP and Nestin in MBENs as compared to DNMBs. Whereas these markers were significantly correlated in MBEN, this was not the case in DNMBs, indicating that there are differences between the astrocytic cell population in these related, but different tumors (**Suppl. Fig. 16e, f**).”

p. 25, l. 1: “Interestingly, we observed that the astrocytic markers GFAP and Nestin were expressed in correlation in MBEN, but not in DNMB. These findings suggest that there may be differences between MBEN and DNMB with regard to the role of astrocytic cells within the tumor microenvironment.”

p. 29, l. 11: “Immunohistochemistry. IHC was conducted on 4-µm thick FFPE tissue sections mounted on adhesive slides followed by drying at 80°C for 15 min. For IHC analysis, a mouse monoclonal GFAP (GA5; Cell signalling) and Nestin (MAB5326; Millipore) antibodies were applied. IHC was performed with an automated immunostainer (Benchmark; Ventana XT) using antigen-retrieval protocol CC1 and a working antibody dilution of 1:2000 for GFAP and 1:200 Nestin with incubation at 37°C for 32 min.”

Is it possible that these tumor astrocytes are radial glia or another type of basal progenitor?

The questions regarding the cell of origin for tumor-derived astrocytic cells is intriguing and we agree with the reviewer that this is of high importance. Since we found the same CNVs and SNVs in the astrocytic-like tumor cells as in the rest of the malignant cells from the same respective cases, it seems likely that they stem from the same cell of origin which for SHH-MB are cerebellar granular neuronal precursors (CGNPs) in the external granular layer (EGL) of the developing cerebellum. This is underlined by the results from our lineage trajectory analysis and by other studies, which have shown that CGNPs can transform into an astrocytic phenotype (Okano-Uchida *et al.*, PNAS, 2004). Based on

our current knowledge, it therefore seems likely that malignant astrocytic cells also stem from early CGNPs in the EGL.

We have added a sentence to the discussion of our results:

p. 24, l. 9: “Based on our current knowledge, it seems likely that malignant astrocytic cells stem from early CGNPs in the EGL.”

Fig. 2 SnRNA-seq reveals a subset of MBEN-cells with similarity to astrocytes.

a UMAP-projection of the integrated dataset (10X snRNA-seq + SMARTseq V2.5 snRNA-seq) shows intratumoral heterogeneity in MBEN. **b** Heatmap depicting marker genes for indicated cell types of all nine clusters. The first five clusters represent malignant MBEN cells. Clusters 6 – 9 are identified as

perivascular fibroblasts/ pericytes, astrocytes, oligodendrocytes/ oligodendroglial precursors and a mixture of microglia and endothelial cells. **c** InferCNV-analysis using non-malignant cells as a control confirms malignant origin for the vast majority of cells as well as a CNV signature of malignant cells in the astrocyte cluster. **d** IGV-representation of case MB266 showing pathogenic SNVs within the gene SMO. Three cells with an astrocytic phenotype harbour the same SNV as detected using bulk sequencing. **e** UMAP projection of the SMARTseq V2.5-dataset without correction for patient batch effects. Cells were mapped to a reference dataset (Sepp et al., bioRxiv 2021) to identify cells with an astrocytic phenotype. As shown, these clustered close to malignant cells. The zoom-in highlights three cells with an astrocytic phenotype and SMO-mutations. **f** UMAP projection as in d. Cells in which the same SNVs as in bulk sequencing were detected are highlighted in red. As shown, three cells within the astrocytic population harbor SNVs. **g** UMAP projection of the 10X snRNA-seq dataset which was not corrected for patient-associated batch effects. Whereas the majority of cells are clustering according to patient, significant mixing of cells occurs in two clusters (encircled), representing non-malignant cells.

Suppl. Fig. 4 Batch effect control and identification of non-malignant cell types as well as astrocytic-like malignant cells based on snRNA-seq derived CNV-analysis.

a Unsupervised clustering of 10X and SMARTseq V2 genotype profiles confirms strong transcriptional similarity between clusters derived from both snRNA-seq methods. **b** Heatmap showing clear correlation between clusters of single cells sequenced with two complementary methods (x-axis: SMARTseq. V2, y-axis: 10X Genomics). **c** UMAP-projection colorized according to patient. No patient-specific batch effects were detected after integration. **d** UMAP-projection colorized according to technology confirms good integration across both technological platforms. **e** Stacked bar chart visualizing the distribution of cells across malignant and non-malignant clusters for every patient. **f** Non-malignant cells were mapped onto an atlas of the developing cerebellum to confirm the designation of non-malignant cell types (monocytes/microglia, astrocytic cells, oligodendrocytes, perivascular and endothelial). **g** Pie chart summarizing the number of astrocytic-like malignant cells per sample. All samples are represented in varying degrees. **h** UMAP-projection of the 10X snRNA-seq dataset without correction for batch effects (also compare: Fig. 3g). Cells from the astrocytic cluster in (f) are highlighted. Whereas one part of the cells with an astrocytic phenotype (blue) cluster with malignant cells, the other part falls into one of the two mixed cluster of non-malignant cells (green). **i** UMAP projection of the SMARTseq V2.5 snRNA-seq. dataset which was not corrected for patient batch effect. Whereas the majority of cells cluster according to patient, significant mixing of cells occurs in one cluster (encircled), representing non-malignant monocytic cells. **j** UMAP projection (SMARTseq V2.5 dataset) in which astrocytic-like cells are highlighted. No cells of the astrocytic cluster (compare f) falls into the mixed, non-malignant cluster.

Suppl. Fig. 16 Immunohistochemistry and gene expression analysis show differences in astrocytic marker expression between MBEN subtypes and DNMBs.

a, b Representative scans of GFAP-stainings in MBEN show expression in the peripheral regions of the nodular compartment, confirming the observations made using spatial transcriptomics. **c** Expression of Nestin is similar to GFAP-expression. **d** Representative example for a GFAP-negative MBEN (subtype)

TCL1). **e, f** The expression of *Nestin* (NES) and *GFAP* shows significant positive correlation in (e) MBEN as compared to (f) DNMB.

Please also see: **Suppl. Tbl. 11**

2. The authors mention that three children in the cohort had relapse. Is there any correlation with specific cell types/signatures and relapse?

Please see answer to question #3

3. Is there any association between age/sex and differentiation of neurons (e.g. more differentiation in younger patients)?

We'd like to thank reviewer #1 for these two interesting questions. In order to test whether any of these three variables (age, sex, and relapse) correlated with specific cell types, the percentage of each cluster per patient was calculated and correlation analysis using Spearman correlates were performed for each cell state with these three variables. No cell state showed a significant positive or negative correlation with age, sex, or relapse. There was only a slight tendency towards a negative correlation between higher age and the percentage of differentiated, neuronal-like tumor cells ($p = 0.2$, $Rho = 0.7$). However, these non-significant results for associations between the number of cells per cell type and clinico-epidemiological variables may be due to the comparably small patient number in our study, including the very small number of older patients.

We have included these findings in our paper:

p. 9, l. 32: "There was no significant association between the percentage of each cluster per patient and either age of onset, sex or relapse status as calculated using Spearman correlations."

p. 28, l. 3: "In order to test for associations between different cell states and the variables age of onset, sex, and relapse status, the percentage of each cell state per patient was calculated. Subsequently, Spearman correlation was used to test for any significant associations."

4. The authors mention a mesenchymal signature, but it is not described in detail in the manuscript.

The term "mesenchymal" was used to refer to non-malignant cells in MBEN which are derived from the mesenchyme. We agree with the reviewer that this term can be misunderstood in the context of (malignant) tumor cells, and we have therefore reframed the respective sentence.

p. 6, l. 35: "Taken together, we found that besides malignant cell subpopulations MBEN contains various non-malignant cells with glial, monocytic, vascular and endothelial transcriptional signatures."

5. For the DoRothEA analysis, how was the measurement of changes in TF activity performed?

We have added a more detailed description on the analysis of TF activity changes in the Methods.

p. 28, l. 16: "Here, we apply a multivariate linear model to every cell in our samples to estimate the log-transformed gene expressions using weights assigned to the interactions between TFs and genes. After the model is trained, the resulting t-values of the slopes serve as scores. A positive score indicates an active pathway, while a negative score indicates an inactive pathway."

6. Can the authors provide more information on microglia/macrophages observed in the snRNA-seq datasets? Are there molecular signatures associated with pro- and anti-inflammatory markers, resident microglia/bone marrow derived macrophage etc?

We thank reviewer #1 for this excellent suggestion. Unfortunately, the number of immune cells and reading depth of the integrated snRNA-seq datasets was not high enough to derive more detailed information on the monocytic populations (microglia, macrophages, dendritic cells) within MBENS. We therefore further analyzed our spatial Molecular Cartography platform transcriptomics data, since this data type demonstrated almost twice higher proportion of non-tumor cells and a number of monocytic marker genes were included in our target gene panel (**Suppl. Tbl. 8**). Here, we found two distinct populations of monocytes, which were primarily distinguished by the expression of *CD16* and *CD163* (**Suppl. Fig. 14**). Whereas *CD16* has been described as a marker of proinflammatory monocytes leaning towards the M1 polarization phenotype, *CD163* is generally seen as a marker of anti-inflammatory monocytes of the M2 polarization phenotype (Jurga *et al.*, 2020). Overall, the ratio of *CD163*⁺- to *CD16*⁺-monocytes was 9:1, hinting at a generally more anti-inflammatory tumor microenvironment. Next, we aimed at determining whether these two subpopulations could be identified as microglia or bone marrow-derived macrophages. To this end, we performed an enrichment analysis of MSiGdb cell type marker genes with subsequent statistical testing using chi-squared tests. Based on the statistically most significant terms, *CD163*⁺ cells mostly resembled microglia. Furthermore, across this population, expression of the microglia marker *TMEM119* was observable (**Suppl. Fig. 14**). The most significant hits for the *CD16*⁺ cluster were associated with dendritic cells and macrophages (**Suppl. Tbl. 10**). However, due to the comparably low number of genes and the limitations of gene enrichment analysis, these results should be seen as exploratory.

These new findings were included in the paper:

p. 18, l. 27: “The monocytic cells could be further subdivided into two distinct populations, which were primarily distinguished by the expression of CD16 and CD163 (**Suppl. Fig. 14c - k**). Whereas CD16 has been described as a marker of proinflammatory monocytes leaning towards the M1 polarization phenotype, CD163 is generally seen as a marker of anti-inflammatory M2 monocytes. Overall, the ration of CD163⁺- to CD16⁺-monocytes was 9:1, hinting at a generally more anti-inflammatory tumor microenvironment (**Suppl. Fig. 14b**). Next, we aimed at determining whether these two subpopulations could be identified as microglia or bone marrow-derived macrophages. To this end, we performed an enrichment analysis of MSiGdb cell type marker genes with subsequent statistical testing using chi-squared tests. Based on the statistically most significant terms, CD163⁺ cells mostly resembled microglia ($p = 4.88 \cdot 10^{-14}$). Furthermore, across this population, weak expression of the microglia marker *TMEM119* was observable (**Suppl. Fig. 14e**). The most significant hits for the CD16⁺ population were associated with dendritic cells ($p = 2.01 \cdot 10^{-14}$) and macrophages ($p = 4.79 \cdot 10^{-14}$) (**Suppl. Tbl. 10**). However, due to the comparably low number of genes and the limitations of gene enrichment analysis, these results remained inconclusive.”

p. 24, l. 30: “Whereas it was not possible to safely distinguish between different subtypes of monocytic cells, we identified two distinct populations that resembled the pro- and anti-inflammatory M1 and M2 phenotype, respectively. Interestingly, the number of anti-inflammatory monocytes was significantly higher.”

Suppl. Fig. 14 Two distinct types of monocytic cells can be distinguished in the MBEN tumor microenvironment.

a UMAP-projection of single cells derived from Molecular Cartography. Two distinct types of monocytes can be distinguished based on *CD16*- and *CD163*-expression. **b** pie chart visualizing the proportions of *CD16*- and *CD163*-positive cells, respectively. **c, d** Features plots of (c) *CD16* and (d) *CD163* expression. **e – k** Feature plots showing the expression of the monocytic marker genes *TMEM116*, *ITGAM*, *AIF1*, *PTPRC*, *CD4*, *CD37*, and *CD14*.

Please also see: **Suppl. Tbl. 8, Suppl. Tbl. 10**

Reviewer #2 (Remarks to the Author): Expert in single-cell and spatial omics, and brain tumours

The current study presents an innovative integration of single cell and spatial transcriptomic analysis of medulloblastoma with extensive nodularity, offering a valuable opportunity to examine spatial dependencies. The manuscript is well-written, though some adaptations to the Figures would enhance interpretation (as noted below). The analysis and interpretations are cutting-edge. Prior to publication, several aspects should be considered:

We would like to thank Reviewer #2 for the favorable evaluation of our manuscript and for his/her valuable suggestions.

1. The CNV alterations in astrocytes were only detected in one patient, and the annotations of patients are missing in the CNV heatmap for the "non-malignant" cells.

As pointed out correctly by reviewer #2, the CNVs that led to our discovery of malignant astrocytic cells were derived from only one patient. This limitation was caused by the fact that CNVs in MBEN are a rare event in an already rare entity, and the thereof resulting difficulty to collect enough cases with available fresh frozen tissue to perform the necessary analyses. As an alternative strategy to derive more evidence for the malignant nature of these cells, we re-analyzed the snRNA-seq data derived using the SMARTseq. V2.5-protocol to look for SNVs in cells with an astrocytic phenotype. We were able to identify mutations in the *SMO* gene, a known SHH-pathway member, in three astrocytic cells from one additional case (**Fig. 2d – f**). The clear copy number variations in astrocytic cells, which we already described in the first version of our manuscript, occurred in another sample (**Fig. 2c**). Thus, these two independent findings further confirm our initial observation that astrocytic cells in MBEN are at least partially of malignant nature and have therefore been included both in **Fig. 2** as well as the text of the respective paragraph. These changes within the manuscript text are highlighted in our answer #2 to a similar question from reviewer #1.

In addition to these new findings, we contacted a number of international collaborators with large biobanks to acquire additional tissues. However, even after several months, we were only able to identify two additional cases with fresh frozen material available for analysis. We used snRNAseq to further process these samples. Unfortunately, upon central neuropathological review, one of these cases turned out to not be an MBEN, whereas the data quality of the other case was too low to retrieve transcriptomes with an acceptable number of genes per nucleus. We therefore were not able to increase the number of cases in our cohort.

As suggested, we also aimed at including patient annotations for non-malignant cells in the upper heatmap of **Fig. 2c**. This turned out as complex, since the relative amounts of non-tumor cells are quite small and vary across samples (**Suppl. Fig 3f,g, Suppl. Fig. 4e**). As shown in **Suppl. Fig. 4e**, each sample contributes to each cluster, and these are combined into a pseudo-bulk dataset that serves as a baseline for CNV calling in all other cells. These are depicted per patient in the lower heatmap. As

reviewer #2 has stated correctly in her/his question, the CNV signature within the astrocytic cluster stems from sample MB295, in which all other cells show the same CNV alteration on chromosome two. In order to confirm these results, we repeated this analysis by separating all astrocytic cells from MB295 and included them as an independent cell population. As shown in the following visualization, this analysis confirmed the clear CNV-signature within this population of cells.

Fig. 1: Visualisation of CNVs for each patient. In this analysis, all astrocytic cells from MB295 were excluded from the reference clusters and introduced as a separate cell population. This analysis confirmed the clear CNV-signature on chr. 2 in these cells, which is identical with the CNVs of all other malignant cells from the same patient (row marked with red arrow).

2. The approach of using non-integrated data to identify tumor cells may be challenging, as batch horizontal integration should remove the confounding factor of technical differences such as the use of two sequencing platforms. Given the significant difference in sequencing depth between the 10X and SMART-seq protocols, it would be insightful to examine the distribution of these platforms across the identified malignant and non-malignant cells.

Bases on this valuable suggestion by reviewer #2, we compared the distribution of different cell types across both platforms and included a new **Suppl. Fig. 3** to illustrate our findings. Both single cell

methods showed similar overall proportions of non-malignant and malignant cells, respectively (Suppl. Fig. 3e). We found more cells with a monocytic phenotype using SMARTseq V2.5, and a higher percentage of vascular/endothelial cells with the 10X protocol. These differences are likely due to technical variation between the two methods, moreover, the comparatively smaller number of overall cells sequenced with SMARTseq V2.5 could also have contributed to this variation (Suppl. Fig. 3f, g).

Suppl. Fig. 3 Quality control and comparability of snRNA-seq using 10X Genomics 3'-V2 and SMARTseq V2.5

a RNA counts per cell for each patient as detected with 10 Genomics 3'-V2 snRNA-seq. **b** Feature counts per cell for each patient as detected with 10 Genomics 3'-V2 snRNA-seq. **c** RNA counts per cell for each patient as detected with SMARTseq V2.5 snRNA-seq. **d** Feature counts per cell for each patient as detected with SMARTseq V2.5 snRNA-seq. **e** Stacked bar chart showing the amount of non-malignant and malignant cells for both methods. **f** Bar chart showing the different non-malignant

cells that were identified using the 10X protocol. **g** Bar chart showing the different non-malignant cells that were identified using the SMARTseq V2.5 protocol.

3. The approach should be quantified through an entropy examination of individual clusters.

Based on this extremely helpful suggestion by reviewer #2, we have performed an entropy examination to test the stability of the individual clusters. For this measurement we used an established quality control method called ROGUE (Liu *et al.*, 2020, Nature Communications). As shown in the following visualization, all derived clusters showed ROGUE-scores around 0.9, indicating internal stability. This result has been included in the Material and Methods section of the paper:

p. 27, l. 36: "This approach was validated using an entropy analysis that showed ROGUE-scores around 0.9 for all clusters, indicating high internal cluster stability."

Fig. II: Barplots visualizing the ROGUE accuracy scores for each cluster of the integrated snRNA-seq. dataset.

4. The term "tumor astrocytes" may not be appropriate, as the scientific literature usually refers to these cells as "tumor-associated astrocytes" or "reactive astrocytes." If malignant cells mirror differentiation trajectories of healthy cells, they are typically described as "astrocytic-like" (Neftel *et al.*, 2019), as the authors have used the terms "CGNP-like" or "neuronal-like."

We agree with reviewer #2 that the term "tumor astrocytes" may be misleading and have adopted his/her proposal to call these cells "astrocytic-like" instead. The wording has been changed throughout the manuscript accordingly and is also used in this rebuttal letter.

5. The hierarchical states analysis using the "sling-shot" approach is not straightforward, and additional pseudotime tools such as STREAM or Monocle2 would provide a clearer demonstration of hierarchical ordered trajectories.

We thank reviewer #2 for this suggestion. Indeed, the Slingshot method has certain limitations with regard to the interpretation of the resulting trajectories, but the benefit of this tool is that it can be

easily adjusted to complex batch effects, which is relevant for our dataset that combines not only different samples, but also two different snRNAseq-methods. For STREAM and Monocle2, this is either not supported or requires specific manual adjustment. In order to avoid batch effects, we used Monocle2 on the sample MB295, which included a large fraction of astrocytic-like malignant cells. The obtained trajectory strongly reflected the main pattern that we observed from Slingshot, with early CGNP-like cells at the start, and neuronally differentiated cells at the end of the trajectory (**Suppl. Fig. 5j – o**). In addition, these findings are in line with the pseudotime trajectories that were performed on an independent set of MBEN-cases by our colleagues Gold *et al.* in their accompanying study. This overarching trajectory reflecting physiological CGNP-development was confirmed when we inspected the relative expression of cerebellar, medulloblastoma and astrocytic marker genes (**Suppl. Fig. 5p – y**). Interestingly, astrocytic-like cells demonstrated a slightly different location within the trajectory as compared to the cluster-based approach using slingshot and were found at the earliest time point prior to the clusters of early, CGNP-like cells. As we have reported in the paper, astrocytic-like cells expressed markers of stem cells, however without showing any significant proliferative activity. Yao *et al.* (Cell, 2020) showed in their murine model that astrocytic-like tumor cells were not stem cells, and in addition Okano-Uchida *et al.* (PNAS, 2004) have shown that CGNPs can differentiate into astroglial cells upon exposition to SHH. We therefore feel that the current knowledge on astrocytic-like cancer cells and CGNP-development support the trajectory derived from slingshot. However, we gladly acknowledge that this remains an open question and that further studies are necessary to conclude on the exact developmental roots of astrocytic-like cells in MBEN and have included these limitations based on the comment by reviewer #2 into the discussion part of our manuscript.

We included these new results in the manuscript:

p. 9, l. 5: “Pseudotemporal ordering using Slingshot50 revealed a continuous lineage starting from the proliferating, early CGNP-like cell states, spanning the intermediate ones and finally congregating in the postmitotic, neuronally differentiated cells (**Fig. 3d**). This lineage resembled physiologic differentiation of CGNPs into normal granular neurons during cerebellar development and was confirmed in an independent cohort of MBEN cases in an accompanying study by Gold *et al.*. Astrocytic-like cells branched off early in the trajectory and prior to the appearance of markers of cell migration (**Fig. 3d**). In addition, we used monocle2 as an additional pseudotime ordering tool on sample MB295, which included a large fraction of astrocytic-like malignant cells. The obtained trajectory strongly reflected the main pattern that we observed from Slingshot, with early CGNP-like cells at the start, and neuronally differentiated cells at the end of the trajectory (**Suppl. Fig. 5j – y**). Interestingly, astrocytic-like cells demonstrated a slightly different location within the trajectory as compared to the cluster-based approach using slingshot and were found at the earliest time point prior to the clusters of early, CGNP-like cells. (**Suppl. Fig. 5j,o**).”

p. 24, l.2: “Whereas the main developmental, CGNP-like trajectory was validated across two methods and datasets, the exact position of astrocytic-like cells within this trajectory was more ambiguous. Since these cells did not show proliferative activity, were located within the bicompartmental MBEN transition zone and based on the findings by Yao *et al.*, who showed in a murine SHH-MB model that astrocytic-like tumor cells were not stem cells, it seems likely that this cell population does not represent a cell of origin for MBEN. However, additional studies – including functional modeling - will be necessary to determine the exact developmental role of this enigmatic cell type.”

p. 27, l.37: “Trajectories were identified the Slingshot and Monocle2 R package as described in the respective studies.”

Suppl. Fig. 5 Malignant cell states differ in their expression of CGNP-marker genes.

a – i UMAP-plots showing the expression of representative marker genes of CGNP development. **a – c** SHH-pathway members HHIP, BOC, and GLI2. **d** epigenetic regulator gene EZH2. **e, f** markers of intermediate CGNP-stages CNTN2 and GRIN2B. **g – i** markers of differentiated CGNPs GABRA6, NRXN3, and RBFOX3. **j** Cell fate trajectory of the sample MB295 as constructed with monocle2. The black arrow shows the direction of the trajectory. **k – o** Projections showing the position of the different cell stages on the MBEN trajectory in sample MB295. The color code is equivalent to (j). **p – y** Visualizations of the relative expression of different marker genes along the MBEN trajectory in MB295.

6. The cell-cell interaction analysis would benefit from the integration of spatial data, such as the cellular components of each spatial niche. For example, the authors later describe a juxtaposition of neuronal-like and migrating CGNP-like cells, and it would be more appropriate to analyze these interactions in accordance with relative spatial distance.

We fully agree with reviewer #2 that the integration of spatial transcriptomic information into the analysis on cell-cell interactions could be very insightful. Based on his/her suggestion, we have investigated whether a tool that could combine the snRNA-seq. ligand-receptor-results with spatial data derived from Molecular Cartography or smRNA-FISH is currently available. Unfortunately, this is not yet the case. The main limitation is the small number of genes (n=12 and n=100, respectively) that these protocols support. We have intensely discussed possibilities to integrate spatial information into our cell-cell interaction analyses, which were performed with the LIANA tool (https://saezlab.github.io/liana/articles/liana_tutorial.html). Currently, a new tool is under development that will allow to perform these analyses but is not available as of yet. However, this is a large and ongoing project that requires a significant body of additional work and will not be finished in the near future. We therefore hope to be able to provide these results in additional, new studies on MBEN.

7. The exclusion of non-malignant cells from the analysis is not clear.

In this study, we aimed at deciphering the developmental trajectories of MBEN in comparison to physiological cerebellar development. Whereas non-malignant cells are likely to play an important role within the tumor microenvironment of MBEN, they are not part of this trajectory and were therefore excluded from the downstream analysis focusing solely on malignant cells. As part of this revision, we have added extensive information on monocytic cells in MBEN and describe different subpopulations of infiltrating monocytes as detected using the Molecular Cartography platform. We found two distinct monocytic populations, which were primarily distinguished by the expression of *CD16* and *CD163*, indicating different biological roles as pro- and anti-inflammatory immune cells within the tumor microenvironment (**Suppl. Fig. 14**). Since reviewer #1 asked specifically about the immune cell infiltration of MBENs, we would kindly refer to our answer to question 6 of reviewer #1 for more information on this topic. We acknowledge that a more detailed investigation of the role of other non-malignant cell types in MBEN is intriguing. All data from this study will be made publicly available to allow other research groups to integrate non-malignant cells from MBEN into their analyses.

8. The integration of single cell data with the microdissection RNA-seq data would improve the cell type distribution analysis, for example using MuSiC or other tools.

Based on this excellent suggestion from reviewer #2, we applied the MuSIC deconvolution tool on the cohort of microdissected MBEN samples using our snRNA-seq data as reference. In this way, we compared the proportion of derived cell states between the nodular and internodular sample groups. From the result it was possible to confirm the specificity of differentiated, neuronal-like cells enriched in nodular and GCNP-like proliferating cells in the internodular compartments as shown in the figure below. This figure was included into the manuscript as **Suppl. Fig. 8c,d** and highlighted as additional confirmation.

Suppl. Fig. 8c,d: Box plots of deconvolution-derived proportions for proliferating, early GCNP-like (c) and differentiated, neuronal like (d) cells as compared between the nodular and internodular compartments. * = p-value < 0.05 (two-sided t-Test). The center line, box limits, whiskers, and points indicate the median, upper/lower quartiles, 1.5× interquartile range and outliers, respectively.

Additionally, these findings have been included in the manuscript as follows:

p. 14, l. 19: “These findings were corroborated by applying the MuSIC-tool to estimate the contributions of differentiated, neuronal-like and proliferating, early GCNP-like cells to the two histological components. We found significant differences that reflected the results of our microdissection-based sequencing analysis (**Suppl. Fig. 8c,d**).”

p. 29, l. 5: The MuSIC deconvolution method was applied on the bulk internodular and nodular RNA-seq datasets using the snRNA-seq derived MBEN cell stages as the reference. Afterwards, the proportions of predicted cell types between groups were compared using two sided t-tests.

9. The Fig. 6 presentation should use consistent colors for cell types and maintain a consistent order of cells in the legend between plots.

We thank reviewer #2 for highlighting the inconsistency within this figure panel. As suggested, we have improved the presentation of Fig. 6 by harmonizing the color code through adapting Fig. 6g and by changing all legends to maintain a consistent order of cells.

10. For a better understanding of cellular co-localization, it would be useful to present integrated data of all three patients in one graph, rather than three individual graphs.

We thank reviewer #2 for this suggestion. The cellular proximity analysis has been repeated for both RNAscope and Molecular Cartography and the updated, integrated visualization is presented in **Fig. 6g** and **Suppl. Fig. 13e**, respectively.

Fig. 6g: Heatmap quantifying the probability of co-localization of cells from each cluster. Red and blue colors indicate high and low probability for co-localization, respectively. (Proliferating) early CGNP-like and stromal/astrocytic cells are co-localizing, whereas early and late stages of the MBEN trajectory are spatially separated.

Suppl. Fig. 13e: Heatmap depicting the integrated cell proximity analysis of all four cases. Cell types with high correlation are more likely to be located next to each other in the tumor microenvironment.

11. The microenvironment with respect to cellular distribution between spatial and single cell appears to differ, and this difference should be quantified and highlighted.

It is correct that the cellular distribution of non-malignant cells varies both within different snRNA-seq techniques and between snRNA-seq and spatial transcriptomic approaches, and we agree with reviewer #2 that these differences deserve further attention. For data derived from Molecular Cartography, the absolute number of cells per cluster has been included in **Suppl. Fig. 12c**. Additionally, in order to facilitate comparisons between the spatial and snRNA-seq data, the relative fractions of each cell type for both methods are visualized in **Suppl. Fig. 12d,e**. Generally, the Molecular Cartography workflow resulted in the detection of approximately twice as many cells from the tumor microenvironment in comparison with snRNA-seq. This was particularly relevant for stromal, endothelial, and immune cells, of which the majority were identified as monocytes. Interestingly, the number of astrocytic cells (non-malignant astrocytes and astrocytic-like tumor cells) was comparable (~2%) across both platforms. Whereas we found a small number of oligodendrocytes using snRNA-seq, they were not identifiable within the spatial transcriptomics data. It is possible that this discrepancy is caused by the limited number of oligodendrocytic marker genes in our gene panel.

We have included these analyses in the revised version of the manuscript:

p. 18, l. 21: "In general, the molecular cartography workflow retrieved larger number of non-malignant cells from the tumor microenvironment, than snRNA-seq (**Suppl. Fig. 12c - e, Suppl. Fig. 14a**). This was particularly relevant for stromal, endothelial, and immune cells, of which the majority were identified as monocytes. Interestingly, the number of astrocytic cells (non-malignant astrocytes and astrocytic-like tumor cells) was comparable across both platforms (**Suppl. Fig. 12d,e**). "

p.25, l. 8: “By comparing snRNA-seq and spatial transcriptomics methods, we found that the number and types of cells from the tumor microenvironment that could be identified with each protocol differed. This was particularly true for monocytic cells, highlighting the need to integrate a spectrum of methods to investigate their role in neuro-oncological diseases.”

Suppl. Fig. 12 Quality control and comparison of non-malignant cell types between Molecular Cartography and snRNA-seq.

a RNA counts per cell for each sample as detected with Molecular Cartography. **b** Number of features per cell for each sample as detected with Molecular Cartography. **c** Bar chart summarizing the absolute number of non-malignant and astrocytic-like malignant cells/ astrocytes as detected with Molecular Cartography. **d, e** Bar charts summarizing the relative abundances of non-malignant cell types and astrocytic-like malignant cells/ astrocytes for (d) Molecular Cartography and (e) snRNA-seq.

12. An extended analysis of cellular co-localization and potential receptor-ligand interactions would offer novel insights on the importance of spatial niches, similarly to the RNA-scope data.

We fully agree that the proposed extended analysis would be beneficial to our study. We therefore tried to perform receptor-ligand-interaction analysis on the spatial transcriptomics dataset obtained with Molecular Cartography. However, due to the restriction of the number of genes included in the panel and the consequent lack of receptor-ligand-pairs, this analysis did not return meaningful results. As pointed out in our answer to question 6, we also currently lack the bioinformatics tools to analyze receptor-ligand-interactions by integrating the more comprehensive snRNA-seq. dataset with spatial transcriptomics. In light of the complete lack of MBEN *in vitro* or *in vivo* models, we currently

see no possibility to perform more comprehensive receptor-ligand-interactions with regard to the MBEN tumor microenvironment. However, we hope to address this and similar questions in a follow up project that will employ the next generation of spatial transcriptomics platforms with larger gene panels.

13. The quality data regarding the UMI per cell and the detected genes per cell for both single cell and spatial data are missing. Information about the 100-plex Molecular Cartography platform, including whether it was customized, is also not available.

We thank Reviewer #2 for highlighting the need to include more detailed data with regard to quality control and experimental setups. As suggested, we have added figures visualizing the UMIs per cell and detected genes per cell for all methods. Data on single nucleus RNA-sequencing can be found in the new **Suppl. Fig. 3a - d**. The data on the spatial transcriptomics platforms using RNAscope and Molecular Cartography has been included as **Suppl. Fig. 10c, d** and **Suppl. Fig. 12a, b**. Furthermore, we have added a **Suppl. Tbl. 8** that lists all genes of the customized Molecular Cartography panel.

Suppl. Fig. 3 Quality control and comparability of snRNA-seq using 10X Genomics 3'-V2 and SMARTseq V2.5

a RNA counts per cell for each patient as detected with 10 Genomics 3'-V2 snRNA-seq. **b** Feature counts per cell for each patient as detected with 10 Genomics 3'-V2 snRNA-seq. **c** RNA counts per cell for each patient as detected with SMARTseq V2.5 snRNA-seq. **d** Feature counts per cell for each patient as detected with SMARTseq V2.5 snRNA-seq. **e** Stacked bar chart showing the amount of non-malignant and malignant cells for both methods. **f** Bar chart showing the different non-malignant cells that were identified using the 10X protocol. **g** Bar chart showing the different non-malignant cells that were identified using the SMARTseq V2.5 protocol.

Suppl. Fig. 10 Quality control smRNA-FISH using RNAScope

a – d Visualizations showing **a** the number of transcripts per single cell, **b** the number of pixels per nucleus, **c** the number of single cells for each sample, and **d** RNA features per cell for each sample. **e** Bar chart showing the number of cells per sample. **f** Principal component analysis showing high concordance between the three cases even without correction, which is further improved by **g** harmony correction. **h, i** Expression of the three marker genes LRRTM4, FOS, and RBFOX3 in the **h** snRNA-seq and **i** smRNA-FISH datasets, respectively. LRRTM4 and FOS were excluded from the further analysis due to strong expression level discrepancies or patient-specific expression (RBFOX3 shown for comparison). **j** Expression of snRNA-seq derived marker genes in the smRNA-FISH dataset.

Suppl. Fig. 12 Quality control and comparison of non-malignant cell types between Molecular Cartography and snRNA-seq.

a RNA counts per cell for each sample as detected with Molecular Cartography. **b** Number of features per cell for each sample as detected with Molecular Cartography.

Please also see **Suppl. Tbl. 8**

Reviewer #3 (Remarks to the Author): Expert in SHH-MB, brain development, and paediatric brain tumours

Medulloblastoma with extensive nodularity (MBEN) is a unique histological subtype of MB characterized by its appearance with distinct compartments. It mainly arises in infants and in very young children, and it is associated with a good prognosis. However, some patients relapse or present disease progression.

In the current study "Compartments in medulloblastoma with extensive nodularity are connected through differentiation along the granular precursor lineage" from Ghasemi et al., the authors investigate the biology underlying the histological compartments of MBEN and how they are interconnected, as it could provide insight into the varied aggressiveness of the disease and the degree of tumor differentiation.

The authors have performed a very thorough analysis using a multimodal transcriptomic approach that improves our understanding of MBEN and would be a fantastic research article for Nature Communications. Their research reveals the intratumoral heterogeneity present within the nodular and internodular compartments, identifying a spectrum of cell stages with a developmental trajectory that mimics the differentiation of CGNP-lineage during cerebellar development, as well as the presence of MBEN-associated tumor astrocytes.

We thank reviewer #3 for his/her positive feedback with regard to our manuscript and for his/her suggestions on how to further improve our study.

Prior to consideration for publication, the following comments should be considered/addressed.

Comments:

Two methods of single nucleus RNA-sequencing were used to analyze the MBEN patient samples: 10X

V2 3'- (n = 9) and the SMARTseq V2.5-protocols (n = 6). These data are later integrated in the analysis. Is there a reason why 3/9 samples are missing or were excluded from the analysis? How did this affect the number of cell populations identified per patient sample? For example, the number of tumor astrocytes (SFig3G)?

MBEN are very rare tumors and the fresh frozen tissue used in this study was derived from archived material at our institution. Unfortunately, for 3/9 samples, the SMARTseq V2.5-protocol failed due to technical issues caused by a malfunctioning pipetting robot, with no material left to repeat the experiments. All of the primary data generated for this study was included and none of it was excluded. As part of this revision, we have prepared two more samples that we obtained from other centers. Unfortunately, one sample turned out to be a DNMB and not an MBEN upon central review, and for the other one the overall tissue quality did not allow to retrieve a meaningful number of cells with sufficient detected genes per cell to be included in this study.

How many CNVs (CNV-signature, line 187) specific to the tumor astrocytes were identified? In how many patients are they identified? In line 195 it is indicated that tumor astrocytes were identified in all samples, however, this is not very clear from the graphs. In Fig 2C it seems that there is one big region with CNV gain in chr2 specific to the tumor astrocytes, but this one is only observed in 1/9 patients (blue) from the panel underneath. Also, in S.Fig 3G there is a pie chart indicating the number of astrocytes per sample suggesting they were identified in all cases although in varied numbers (min 3, max 208). However, in Figs D-E, F-G it seems that they are only present for some patients.

Reviewer #3 raises a very important point. We discovered the presence of malignant astrocytic-like tumor cells based on the clear CNV signature on chromosome 2 that could be found in all cells from sample MB295, including all astrocytic-like cells from this case. We therefore didn't identify CNVs specific to astrocytic-like tumor cells, but instead used the presence of a CNV signature to identify these cells within the astrocytic cluster. The numbers in the pie chart in Suppl. Fig. 3g (now **Suppl. Fig. 4g**) were based on our strategy to approximate which astrocytic cells would be non-malignant and malignant, respectively. We observed that astrocytic cells clustered either in a mixed cluster of non-malignant or within patient-specific malignant populations when analyzed without correcting for patient-specific batch effects, and in the absence of any form of *in vitro* or *in vivo* MBEN models, we used these findings to estimate which cells were non-malignant and malignant, respectively (**Fig. 2g**, **Suppl. Fig. 4h - j**). This approach has been used in a number of published studies to distinguish between non-malignant and malignant cells (for instance: Patel *et al.*, 2014, Science, Venteicher *et al.*, 2017, Science, Filbin *et al.*, 2018, Science). As we acknowledge in the discussion of our results, this strategy does not allow to distinguish fully between these two highly similar cell populations and to unravel the functional and genomic differences between them. Unfortunately, a suitable alternative to this bioinformatically rooted strategy does not exist due to the complete lack of faithful *in vitro* or *in vivo* models of MBEN.

As an alternative strategy to derive more evidence for the malignant nature of these cells, we re-analyzed the snRNA-seq data derived using the SMARTseq. V2.5-protocol to look for SNVs in cells with an astrocytic phenotype. We were able to identify mutations in the *SMO* gene, a known SHH-pathway member, in three astrocytic cells from one additional case (**Fig. 2d - f**). The clear copy number variations in astrocytic cells, which we already described in the first version of our manuscript, occurred in another sample (**Fig. 2c**). Thus, these two independent findings further confirm our initial observation that astrocytic cells in MBEN are at least partially of malignant nature and have therefore

been included both in **Fig. 2** as well as the text of the respective paragraph. These changes are listed and highlighted in our answer #2 to a similar question from reviewer #1.

In addition, we put significant efforts into increasing our cohort size through acquiring more MBENs with CNV that would allow to analyze new samples in which astrocytic-like malignant cells would be faithfully identifiable. Unfortunately, these attempts were not successful due to the extreme rarity of these cases. For further information with regard to the acquisition of new samples, we would kindly refer to our reply to the similar question #1 from reviewer #2.

A Table with the list of CNVs (CNV-signature) specific to the tumor astrocytes should be provided, particularly indicating which ones were identified for each patient.

The suggestion that malignant, astrocytic-like cells could show distinct CNV-profiles is very intriguing. As pointed out in our previous reply, we did not observe CNVs specific to astrocytic-like malignant cells but used the CNV signature from one patient to identify the presence of these cells within the general compartment of astrocytic cells. This approach was verified by analyzing astrocytic cells from the respective case MB295 separately, as shown in Fig. I in this letter (answer to question 1 from reviewer #2). Our findings of SNVs in astrocytic cells from another patient may serve as a confirmation of these initial results.

Line 254. How was the pseudo-temporal ordering performed?

The pseudotime analysis was performed using the slingshot-algorithm (Street *et al.*, BMC Genomics, 2018), which constructs a minimum-spanning tree based on the previously established clusters and has been shown to be stable with regard to varying batch effects, which was important for our study due to the fact that we combined both different patients and two methods of snRNA-seq. The main graph from early CGNP-like to neuronally differentiated-like malignant cells was verified in the independent MBEN-dataset of the accompanying study by Gold *et al.* Based on the reviewer's comments, we also performed a pseudotime analysis on one sample with a large number of astrocytic-like MBEN cells using another algorithm (Monocle 2) which fully confirmed the main developmental trajectory (**Suppl. Fig. 5j – y**). Interestingly, astrocytic-like cells demonstrated a slightly different location within the trajectory as compared to the cluster-based approach using slingshot and were found at the earliest time point prior to the clusters of early, CGNP-like cells. As we have reported in the paper, astrocytic-like cells expressed markers of stem cells, however without showing any significant proliferative activity. Yao *et al.* (Cell, 2020) showed in their murine model that astrocytic-like tumor cells were not stem cells, and in addition Okano-Uchida *et al.* (PNAS, 2004) have shown that CGNPs can differentiate into astro-glial cells upon exposition to SHH. We therefore feel that the current knowledge on astrocytic-like cancer cells and CGNP-development support the trajectory derived from slingshot. However, we gladly acknowledge that this remains an open question and that further studies are necessary to conclude on the exact developmental roots of astrocytic-like cells in MBEN and have included these limitations into the discussion part of our manuscript. We kindly refer to our reply to question #5 by reviewer #2 for the new Suppl. Fig. 5 and the respective updates in the manuscript.

Fig 4. Could some of the receptor-ligand-interactions identified be validated?

The ligand-receptor-interaction method that we used (i.e. LIANA) already takes into account and integrates six different bioinformatics tools for the analysis of ligand-receptor-pairs. Therefore, the resulting predictions are already validated *in silico* by reporting the consensus of these methods. We

acknowledge that this analysis workflow was not entirely clear based on the current information given in the material and methods section. Thus, we have added the following, more detailed description in the respective paragraph of the manuscript:

p. 28, l. 23: “In order to analyze cell-cell communication between cell types, we conducted a ligand-receptor analysis with the Ligand-receptor Analysis framework v0.1.6 (LIANA). LIANA combines different prior knowledge resources and different in silico methods by taking the consensus of the ligand-receptor predictions. The tool was used with default settings as described in the tutorial at https://saezlab.github.io/liana/articles/liana_tutorial.html. This results in the usage of the methods CellPhoneDB, Connectome, log2FC, NATMI, SingleCellSignalR and CellChat. A consensus between these methods was reached by applying the robust rank aggregate method. Five expertly curated CCC resources derived from the Omnipath database, namely CellPhoneDB, CellChat, ICELLNET, connectomeDB2020 and CellTalkDB, were used as prior knowledge.”

Fig 5 A. The cut off (logFC, adjusted p-value) used to identify the subset of differentially expressed genes among nodular and internodular compartments should be indicated.

As suggested, we have included the filtering controls for the DEGs into the corresponding Methods section:

p. 28, l. 38: “Differentially expressed genes between nodular and inter-nodular blocks were detected via the limma package, with considering the tumor sample for batch effect correction and using minimum limits of 0.5 for log2 fold change, and 0.05 for adjusted p-values.”

Fig 5 B and C. Statistical analysis comparing the expression levels of TMEM108 and TRIM9 should be included.

The exact p-values have been included in **Fig. 5b,c** and the corresponding figure legend:

“**b** TMEM108, a marker of the internodular compartment, is mainly expressed in early CGNP-like cells. Feature plot on the left depicting gene expression in the snRNA-seq dataset. Boxplot on the right showing expression (measured in batch-effect adjusted RPKM) in microdissected MBEN tissue (limma adjusted p-val = 0.02). **c** TRIM9, a marker of the nodular compartment, is mainly expressed in later stages of MBEN development. Feature plot on the left depicting gene expression in the snRNA-seq dataset. Box plot on the right showing expression in microdissected MBEN tissue (limma adjusted p-val: 0.03).”

Spatial transcriptomics correlates cell stages in MBEN to its histologic compartments. “For RNAscope, twelve genes were chosen as markers for the different snRNA-seq clusters, and also considering biological significance”. How were these genes chosen? S. Tbl. 7 indicates the gene probes used for the 12 genes.

The majority of the 12 genes for smRNA-FISH (RNAscope) were selected as marker genes for cell state populations. These were either derived from our snRNAseq data, for instance *DIAPH3* for proliferating, early CGNP-like and *NRXN3* for differentiated neuronal-like tumor cells, or from the literature, for instance *CNTN2* for migrating CGNP-like cells. In addition, *PTCH1* and *GLI1* were chosen as marker genes for the SHH-pathway, and *FOS* as an overall marker for immune cells. The expression of these marker genes in the snRNA-seq and smRNA-FISH datasets is shown in **Fig. 6a, b**.

Additional information should be included indicating: from which population were each chosen, the

reason why they were included (explanation and biological significance +/- highest expression and unique to the cluster?) and the 2 genes that were excluded.

As suggested, we have extensively revised **Suppl. Tbl. 7** and included more detailed information on the genes in our smRNA-FISH panel. The genes *RBFOX3*, *DIAPH3*, *CNTN2*, *NRXN3*, *LAMA2*, *PBX3*, *PLEKHA7*, *TRPM3*, *LRRTM4*, and *FOS* were chosen as marker genes that were differentially expressed in the snRNA-seq clusters. Whereas some markers are exclusive to one cluster, we also aimed at choosing some markers that covered neighboring cell stages of the MBEN developmental trajectory to also cover transitional stages and to have some redundancy in case of technological issues. In addition, the two genes *GLI1* and *PTCH1* were chosen as marker genes of the *SHH*-pathway. Furthermore, and in addition to **Fig. 6a,b**, which show the expression of these genes in both the snRNA-seq and the smRNA-FISH datasets, we'd like to draw attention towards the quality control data and more detailed information on the methods included in **Suppl. Fig. 10**. As shown in **Suppl. Fig. 10h, j**, the gene *LRRTM4* was excluded due to being specifically expressed in only one patient (MB299) in the spatial transcriptomic experiments. The inclusion of this marker would have introduced a significant patient-specific bias into our analysis. The second excluded gene was *FOS*, which was originally intended as a general marker gene for immune cells. Unfortunately, it turned out to show very significant expression level discrepancies in comparison to our snRNA-seq dataset. In order to overcome these limitations, we then used a broader range of immune cell marker genes as part of our panel for Molecular Cartography, which allowed to safely identify immune cells in MBEN.

Suppl. Fig. 10 Quality control smRNA-FISH using RNAscope

a – d Visualizations showing **a** the number of transcripts per single cell, **b** the number of pixels per nucleus, **c** the number of single cells for each sample, and **d** RNA features per cell for each sample. **e** Bar chart showing the number of cells per sample. **f** Principal component analysis showing high concordance between the three cases even without correction, which is further improved by **g** harmony correction. **h, i** Expression of the three marker genes *LRRTM4*, *FOS*, and *RBFOX3* in the **h** snRNA-seq and **i** smRNA-FISH datasets, respectively. *LRRTM4* and *FOS* were excluded from the further analysis due to strong expression level discrepancies or patient-specific expression (*RBFOX3* shown for comparison).

The authors hypothesize about the clinical relevance of their findings. Additional analysis could be considered, including:

How could this finding be translated into the clinical setting? From the cohort of 26 samples analyzed (microdissected tissue), IHC with the 2 set of genes representative of each compartment could be performed (TMEM108/TRIM9; NRX3/LAMA2).

Does it have prognostic implications? The proportion of differentiated vs non-differentiated populations could be calculated for the cohort of 26 samples analyzed and integrate the results with survival analysis. What are the differences between patients with a better outcome compared to those that progress or are metastatic?

We thank reviewer #3 for these excellent suggestions and agree that the possibility of translating our findings into the clinic is intriguing. Unfortunately, there was not enough FFPE-derived tissue left to perform extensive IHC stains in our in-house MBEN cohort, since this material had already been used for bulk sequencing, the microdissection experiments and additional IHC stains for GFAP and Nestin as part of this revision. Instead, we took several different approaches to investigate for associations between the different MBEN cell stages and epidemiological or clinical variables:

Firstly, and as explained in detail in our reply to the similar questions #2 and #3 by reviewer #1, we searched for correlations between the abundance of each cell state and the variables age, sex, and relapse status within our cohort of nine patients. To this end, the percentage of each cluster per patient was calculated and correlation analysis using Spearman correlates were performed for each cell state with these three variables. No cell state showed a significant positive or negative correlation with age, sex, or relapse.

Secondly, we performed a multigene survival analysis for an extended cohort of MBEN with available bulk RNA-sequencing and clinical data (n=39) (Korshunov *et al.*, 2020) to check which genes' expression would correlate with progression free survival (PFS). We then systematically compared a list of 106 genes that were either associated with favorable or unfavorable PFS with all genes that were differentially expressed in each MBEN cluster. Amongst the genes that were differentially expressed in different MBEN cell stages, the candidates *RAB3C*, *FAM153A*, *STOX2*, *SLCO1C1*, and *GAN* were associated with a low risk of progression, whereas the genes *RPL7A*, *RPS8*, *RPLP2*, *RPL35*, *RPL32*, *RPS7* and *ASAP2* showed correlation with a high risk of recurring disease. Upon closer inspection, low-risk genes tended to be expressed in later cell stages, whereas high-risk genes were almost all coding for ribosomal proteins and mostly expressed in proliferating cells and cells with an upregulated stress response. Generally, all associations were weak as well as not specific for single clusters (**Fig. III**) and would therefore not allow to use any of these genes as a clinical marker.

Thirdly, we used the same MBEN cohort (n = 39) to run deconvolution using the MuSiC tool with the corresponding snRNA-seq-derived tumor clusters as the reference. In this way, we inspected the

impact of obtained deconvolution proportions of cell types per sample in association with survival. However, the result did not demonstrate any statistically significant associations.

Taken together, we did not find any reliable associations between the abundance of cell states and clinical survival in MBEN. These results could be caused by the limited number of relapses and deaths within our cohort, pointing towards the fact that larger data sets are likely necessary to detect more subtle, clinically relevant associations. For the moment, we feel that any claims with regard to transcriptional biomarkers for survival based on our study would not be supported significantly enough by the data. Therefore, we would like to refrain from including these analyses in our manuscript.

Fig. III: Expression of clinical low- and high-risk gene markers derived from bulk-RNA sequencing is unspecific and not restricted to single clusters.

Data availability: Data should be made publicly available, and a data availability statement should be included.

We can confirm that all data in this study will be made publicly available. We have already uploaded the snRNA-seq data and the bulk RNA- and DNA panel-seq data to GEO. Furthermore, the data from our spatial transcriptomics experiments have been uploaded to the BioImage Archive both in the

form of raw images and after the initial processing including cell segmentation. In addition, the script for processing this data has been published via GitHub.

We have included a statement on data availability:

p. 32, l. 26: "Data availability

The snRNA-seq and bulk-sequencing (RNA, microdissected) data are deposited in GEO database and available under the combined accession number GSE239854. All raw images and processed data after cell segmentation from spatial transcriptomics experiments have been deposited at BioImage Archive and can be accessed under the accession numbers S-BIAD825, S-BIAD826.

Code availability

Scripts for processing the raw data and generating the figures are available via GitHub repository: https://github.com/kokonech/MBEN_snData_analysis."

Minor comments:

Fig. 1 Method spectrum and molecular features of the MBEN patient cohort.

- Fig 1B. Labelling of axis is missing for the t-SNE plot

We have added axis labels to the t-SNE plot.

- Fig 1C. It is not very clear how many patients present the alteration. The X axis indicates patients 1-5 for the cohort of 9, and there are 2 squares to indicate 1 patient. Is the x axis numbering wrong?

We acknowledge that the presentation of alterations in the SHH-pathway was not intuitive. Therefore, we have replaced the original diagram with a pie chart.

- Fig 1D. Scale magnification should be included. It would also be helpful that an example of each compartment is highlighted for the non-experts in histopathology.

We apologize for the insufficient image quality of Fig. 1d. We have decided to replace this scan with another representative HE staining of an MBEN which includes a scale bar and makes it easier to appreciate the two distinct histological compartments of this tumor. As suggested, we have highlighted both components.

Fig. 1 Method spectrum and molecular features of the MBEN patient cohort.

a Visual summary of all methods that were applied to investigate the MBEN cohort. **b** t-SNE depicting a clustering of the MBEN cohort (n = 9) with a reference cohort of 391 molecularly characterized MBs. All nine samples clustered with the SHH-subtype. **c** Bar chartPie chart summarizing the alterations in the SHH-pathway in both blood and tumor that were detected using DNA panel seq. **d** Representative H.E. staining showing the two characteristic histological components of MBEN. The black asterisk marks the internodular component, whereas the black arrow points towards a tumor nodule. **e** Representative coronal MRI image in an MBEN patient showing a large cerebellar mass with grapevine-like appearance.

Fig. 2 SnRNA-seq reveals a subset of MBEN-cells with similarity to astrocytes.

- Fig. 2C. It is a bit confusing to understand whether the colors represent patient samples or cell populations. The legend corresponding to the top panel should be moved up so it is clear that the first panel focusses on cell populations while the second one on patient samples.

- Fig. 2E. Letter E is duplicated in the Fig. legend, one is C.

We have moved the first legend up as suggested, and the duplicated letter has been replaced.

Fig4. Biological processes that regulate MBEN differentiation mimic CGNP development.

- specify which population each cluster represents

We have added the populations names to each cluster in the UMAP plot of **Fig. 4C**.

Fig 6. Spatial transcriptomics map developmentally distinct cell stages to spatially distinct tumor compartments.

- Add scale in tissue sections

Throughout the paper, scale bars have been added to all tissue sections or spatial transcriptomic images.

Fig 8. MBEN histology recapitulates physiological CGNP development.

- Missing Fig. legend

A legend has been added to Fig. 8.

References

- Format issue in reference 56, it includes 5 weblinks.

We have corrected the format issue in the respective reference.

Additional edits in the revised manuscript:

- **Abstract:** We have shortened the abstract to account for the word limit of 150 words.
- **Fig. 2:** The cell population formerly called “Fibroblast/ perivascular cells” has been renamed into “perivascular/ stromal” to harmonize the cluster designations with the names used in the description of the spatial transcriptomics data.
- **Material and Methods:** On p. 32, l. 12 a sentence has been added to the paragraph on Molecular Cartography (Resolve Biosciences) to give more detailed information on the respective downstream analysis: “Cell states and types were annotated by comparison to the snRNA-seq dataset and by visual inspection of marker genes. Clusters which only differed marginally in gene expression were combined.”
- **Acknowledgments:** We have added a dedication to Prof. Felice Giangaspero, a dear colleague and co-author on this paper, who tragically passed away during the course of this revision.
- **Figures and legends:** In accordance with the journal’s formatting guidelines, capital letters have been lower case letters in all figures and legends to number figure panels.
- **References:** Where necessary, additional references have been quoted. These are not highlighted in the revised manuscript, since the inclusion of references in track changes can cause severe formatting issues when using automated citation manager programs, such as in this case Endnote.

Reviewers' Comments:

Reviewer #1:

Remarks to the Author:

The authors have addressed my concerns and have significantly improved the manuscript. I recommend publication of this study in Nature Communications.

Reviewer #2:

Remarks to the Author:

The authors have done a great job to revise the manuscript. My concerns were sufficiently addressed, hope to see the work published soon !

Reviewer #3:

Remarks to the Author:

The authors have undertaken extensive and comprehensive revisions to their manuscript, and all of my concerns have thus been addressed.